# Interpreting and Controlling LLM Reasoning through Integrated Policy Gradient

## Abstract

Large language models (LLMs) demonstrate strong reasoning abilities in solving complex real-world problems. Yet, the internal mechanisms driving these complex reasoning behaviors remain opaque, raising concerns regarding truthfulness, safety, and controllability in practical applications. Existing interpretability approaches targeting reasoning either rely on human-annotated contrastive pairs to derive control vectors, which limits reliability and generalization, or identify neurons correlated with superficial textual concepts, failing to capture the complexity of reasoning processes. Consequently, current methods struggle to precisely localize complex reasoning mechanisms or capture causal effects from model internal workings to the reasoning outputs. In this paper, we build on causality-aware and outcome-oriented principles that focus on identifying components that have causal contributions to reasoning behavior where outcomes are cumulated by long-range effects. We propose Integrated Policy Gradient (IPG), a novel framework that attributes reasoning behaviors to model inner workings like neurons, by propagating compound outcome-based signals (e.g., post reasoning accuracy) backward through model inference trajectories. IPG is efficient requiring only a few calls to the standard gradient operator, which uncovers causal structures governing complex reasoning and avoids large manual supervision. Empirical evaluations demonstrate that our approach achieves more precise mechanistic interpretability and enables reliable modulation of reasoning behaviors across diverse reasoning models.

## 1 Introduction

Large Language Models (LLMs) have shown outstanding performance in addressing natural language processing (NLP) tasks (Touvron et al., 2023; Brown et al., 2020; Qwen Team, 2024). Beyond simple next-token prediction, modern LLMs now demonstrate sophisticated reasoning abilities, including structured, step-by-step problem solving (Wei et al., 2022; Khot et al., 2023). Despite these advancements, the internal reasoning mechanisms underlying large language models, especially how they represent and encode such capability, remain unclear. This lack of transparency poses challenges for truthfulness, safety, and controllability in reasoning systems (Wang et al., 2025b).

Mechanistic interpretability studies suggest that the internal representation space of LLMs encodes reasoning-related concepts. Existing interpretability approaches for reasoning fall into two categories. The first derives control vectors in the representation space to induce desired behaviors (Højer et al., 2025; Wang et al., 2025a), but these methods heavily rely on well-designed contrastive input pairs, raising uncertainty about whether the target reasoning behaviors are truly elicited and limiting their reliability in coupling behaviors to specific prompts. The second line of work analyzes individual internal components, such as neurons (Stolfo et al., 2023; Rai & Yao, 2024) or sparse features (Galichin et al., 2025), focusing on mediation analysis (Pearl, 2001) on short-term effect, e.g., arithmetic, or on correlations between their activations and text patterns, e.g., reasoning-related tokens. While informative, these prevalent approaches face one or both of the following limitations: (i) *Lack of causality*: correlation-based analyses fail to isolate mechanisms that causally drive reasoning behaviors. (ii) *Limited scope*: they capture only short-term effects of the inner workings, neglecting their cumulative influence across multi-step reasoning (Sui et al., 2025), such as on final task accuracy.

We design our approach for mechanistic interpretability of reasoning in LLMs based on two key principles: *causality-aware* and *outcome-oriented*. The effects of internal components should be evaluated by their causal contribution to reasoning behaviors, and these contributions should be measured with respect to the final outcome of reasoning. Since reasoning unfolds over long horizons, the outcome (e.g., correctness of the final answer) captures the integrated influence of intermediate steps (Sui et al., 2025). Both principles further enable effective steering of LLM reasoning.

Building on these principles, we propose Integrated Policy Gradient (IPG), a novel training-free framework for interpreting and controlling reasoning in LLMs. IPG first applies gradient-based attribution to identify influential internal *components* (such as neuron or sparse features) within the LLM model that causally shape reasoning behavior. However, standard gradient approaches are limited because reasoning outcomes are typically non-differentiable, e.g., final answer correctness provided by external verifier (Guo et al., 2025). To address this, IPG incorporates policy gradient (Mnih et al., 2016; Schulman et al., 2017), which propagates outcome-based reward signals backward through the entire inference trajectory. This enables attribution of cumulative, long-range effects, overcoming challenges of sparse and non-differentiable outcome signals. Inspired by Sundararajan et al. (2017); Dhamdhere et al. (2019), IPG further aggregates policy gradients along the path from a baseline to the observed activation level of each component. This integrated path accumulation ensures completeness and faithfulness in attributing reasoning behaviors. Importantly, IPG achieves this efficiently, requiring only a small number of gradient computations without updating model parameters. By eliminating the need for hand-crafted contrastive input pairs and moving beyond correlation with surface text patterns, IPG provides a principled, causality-aware, and outcome-oriented approach to mechanistic interpretability of LLM reasoning. Figure 1 illustrates the paradigms of current methods and our IPG and shows our advantages by the effectiveness of controlling reasoning behavior.

We evaluate the reasoning-related components identified by IPG on two representative open-source LLMs and demonstrate their superior *control effectiveness* in steering reasoning behaviors. Across four large-scale reasoning benchmarks, IPG consistently outperforms prior approaches. Moreover, the identified components exhibit strong *transferability*, results derived from prompt-elicited reasoning models (Yang et al., 2024) can be directly applied to training-induced variants (Guo et al., 2025) within the same model family, eliminating redundant interpretation efforts. Finally, individual components, such as specific neurons, enable *fine-grained controllability* over distinct aspects of reasoning, including problem comprehension and task decomposition, underscoring the potential of IPG for advancing mechanistic interpretability of LLM reasoning.

In summary, our work makes three key contributions, centered on improving the mechanistic understanding and controllability of reasoning in large language models. (i) We introduce Integrated Policy Gradient (IPG), a novel training-free framework towards reasoning mechanism that overcomes the limitations of prior approaches by moving beyond correlation analysis and hand-crafted contrastive inputs. IPG is both causality-aware and outcome-oriented, attributing reasoning behaviors directly to internal components based on long-horizon outcomes. (ii) We demonstrate that IPG achieves superior control effectiveness, enabling not only reliable steering of reasoning behaviors but also fine-grained controllability over specific reasoning skills. (iii) We show that the identified components exhibit strong transferability across model variants, substantially reducing redundant interpretation efforts. Together, these contributions advance the mechanistic understanding and controllability of reasoning in large language models.

## 2 RELATED WORK

**Reasoning in Large Language Models.** Reasoning has become a core capability of Large Language Models (LLMs). To improve LLM's reasoning ability, existing methods primarily include prompt-elicited (Wei et al., 2022; Khot et al., 2023) and training-induced reasoning. The training-induced methods involve reinforcement learning (RL) (Guo et al., 2025; OpenAI, 2024; Qwen Team, 2024), optimizing reasoning behaviors with reward signal. Alternatively, other methods utilize supervised fine-tuning, where models are explicitly trained on datasets of reasoning chains (Yue et al., 2023) or distilled from more powerful reasoning models (Guo et al., 2025). Despite the success in enabling reasoning capabilities, how these models encode their internal reasoning remains unclear.

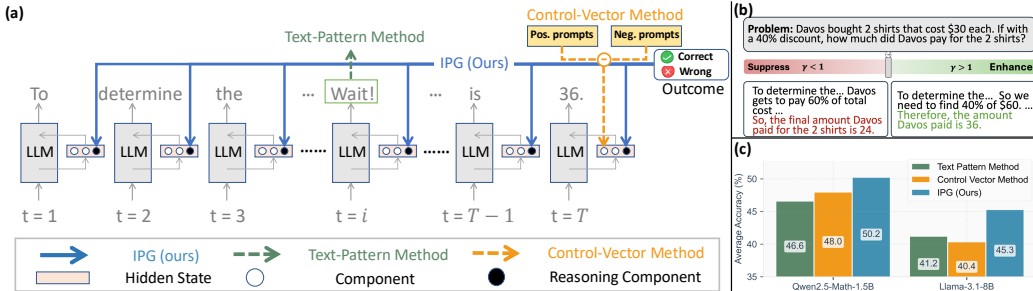

Figure 1: Paradigms of interpretability for reasoning in LLMs and comparison results on reasoning datasets. **(a)** Illustration of the three paradigms, regarding (i) text-pattern methods that associate high activations with special tokens (e.g., "Wait"); (ii) control-vector methods that rely on human-craft contrastive input pairs; and (iii) our proposed Integrated Policy Gradient method. $t$ is the reasoning time step. **(b)** Intervention procedure of our method: selected components are scaled by a factor $\gamma$ (see Section 3.3). **(c)** Average accuracy after steering across reasoning tasks for each paradigm.

**Mechanistic Interpretability in Language Models for Reasoning.** Mechanistic interpretability seeks to reverse-engineer the computations of language models by examining their internal representations (Bereska & Gavves, 2024). Reasoning, however, requires multi-step and long-horizon internal computation, posing unique challenges for the prevalent MI method (Plaat et al., 2025). Representation space steering aims to control model behavior, by adding vectors derived from contrastive input-output pairs (Højer et al., 2025; Zhu et al., 2025). These methods require carefully annotated contrastive pairs and prompt design, which struggle to ensure the intended target states are truly elicited and makes it challenging to reliably align the reasoning behaviors (Højer et al., 2025). Other work targets concept-specific neurons (Dai et al., 2022; Gurnee et al., 2023; Wang et al., 2022; 2025c), and feature extractions using sparse autoencoders (Gao et al., 2025; Huben et al., 2024). Such methods have also been applied to reasoning-related concepts, including arithmetic (Rai & Yao, 2024) neurons and reason-specific features (Galichin et al., 2025), focusing on correlating activations with surface text patterns (e.g., reasoning tokens like "wait"). Intervention-based methods like Causal Mediation Analysis (CMA) (Stolfo et al., 2023) can identify causal components in reasoning, but they primarily focus on short-term tasks such as simple arithmetic, limiting their applicability to the dynamic, multi-step nature of reasoning. Overall, existing methods do not establish cause relation between target reasoning behavior and components, or struggle to capture cumulative, multi-step influences in reasoning. In contrast, our approach is causality-aware and outcome-oriented, attributing reasoning behavior without relying on human annotations or hand-crafted text patterns.

## 3 INTEGRATED POLICY GRADIENT (IPG) METHOD

In this section, we introduce IPG, a framework for identifying internal components that causally underlie reasoning in LLMs and for exerting fine-grained control over their reasoning behavior, depicted in part (a) of Figure 1. Using outcome-aware reward signals (Section 3.1), IPG attributes reasoning performance to internal activations via gradient-based attributions (Section 3.2). The selected components are then scaled to modulate reasoning behavior (Section 3.3), enabling interpretable and causal interventions.

### 3.1 REASONING BEHAVIOR MEASUREMENT

Complex reasoning abilities that unfold over long-horizon inference trajectories have become a central focus in recent language model research (Guo et al., 2025). A key challenge for mechanistic interpretability is to identify internal components that *causally* drive reasoning behavior, where the final outcome rely on multi-step dependencies across the trajectory.

To this end, we employ gradient-based attribution methods to localize hidden components that cause reasoning outcomes. However, standard gradient approaches are limited in this setting, since reasoning feedback is typically sparse and non-differentiable, and the objective (e.g., final-answer correctness) is often only realized after an entire inference sequence. To overcome this limitation, we integrate policy-gradient methods (Mnih et al., 2016; Schulman et al., 2017), which propagate

outcome-aware signals (such as correctness rewards) backward through the trajectory, thereby enabling attribution over long-horizon reasoning effects.

To obtain reliable measurements of reasoning behavior, we extend the perspective of policy gradient (Mnih et al., 2016; Schulman et al., 2017) from parameter space to representation space by propagating outcome-based signals back to intermediate hidden states $\mathbf{h}$. Concretely, for a measure $J(\mathbf{h})$ that assesses reasoning ability, the policy gradient with respect to the hidden state is

$$\frac{\partial J(\mathbf{h})}{\partial \mathbf{h}} = \mathbb{E}_{\tau \sim \pi_\theta} \left[ \sum_{t=1}^{T} \frac{\partial}{\partial \mathbf{h}} \log \pi_\theta \big( a_t \mid s_t; \mathbf{h} \big) \cdot A^\pi(s_t, a_t) \right] , \qquad (1)$$

where $\tau = (a_1, \ldots, a_T)$ is a reasoning trajectory following $\pi_\theta$ which is the policy parameterized by $\theta$, i.e., the LLM to be interpreted. $s_t$ is the prefix tokens and $a_t$ is the next token at the reasoning time step $t$. $A^\pi(s_t, a_t)$ is the advantage function estimating the long-horizon benefit of taking action $a_t$ and $s_t$. In practice, it can be estimated using GRPO (Shao et al., 2024), that measures the relative benefit of taking action $a_t$ at state $s_t$ compared to the average. See Appendix A.3 for details. The targeted hidden representation $\mathbf{h}$ is treated as the variable of interest. The process is depicted in part (a) of Figure 1.

Rather than using policy gradients to update model parameters $\theta$, we compute gradients with respect to $\mathbf{h}$ to obtain attribution signals that reflect the long-horizon impact of internal activations on reasoning outcomes. These signals form the basis for quantifying their effects on reasoning performance and selecting components for subsequent causal interventions (Sections 3.2 and 3.3).

## 3.2 Reasoning Components Attribution

Next, we attribute outcome-aware reasoning signals to internal hidden components. Prior interpretability methods often rely on contrastive example pairs or hand-crafted input patterns (e.g., arithmetic templates) (Højer et al., 2025; Rai & Yao, 2024; Galichin et al., 2025). Such strategies are limited: (i) they do not establish *causal* links between components and long-horizon reasoning outcomes, and (ii) they fail to capture the cumulative, multi-step dependencies that produce the final reasoning result. Comparisons between these paradigms are shown in part (a) of Figure 1.

To address these limitations, we propose Integrated Policy Gradient (IPG), based on two principles: *causality-aware* and *outcome-oriented*. This means that the components identified should causally contribute the reasoning behavior and affects long-horizon reasoning outcomes. Given a hidden state $\mathbf{h} = [h_1, \ldots, h_i, \ldots, h_n] \in \mathbb{R}^n$ with hidden dimension $n$ in a language model layer corresponding to an input $\mathbf{x}$. Our goal is to measure the influence of each component $h_i \in \mathbf{h}$ on the model's reasoning ability. We define the attribution score for reasoning ability with respect to each component $h_i$ as

$$\text{IPG}(h_i; \mathbf{x}) = (h_i - h_i') \int_0^1 \mathbb{E}_{\tau \sim \pi_\theta} \left[ \sum_{t=1}^{T} \frac{\partial}{\partial h_i} \log \pi_\theta(a_t \mid s_t; h' + \alpha(h - h')) \cdot A^\pi(s_t, a_t) \right] d\alpha ,$$
$$(2)$$

where $h_i'$ is the relative baseline value of $h_i$. Deriving from Equation 1, for each generated trajectory $\tau = (a_1, \ldots, a_T)$, $\log \pi_\theta \big( a_t \mid s_t; h' + \alpha(h - h') \big)$ is the log-probability of selecting token $a_t$ given the context $s_t$ under the interpolated hidden state, it is weighted by the advantage $A^\pi(s_t, a_t)$, making the resulting gradients outcome-relevant (Zhong et al., 2025). We combine gradient-based attribution with policy-gradient methods (Mnih et al., 2016; Schulman et al., 2017), so that reward signals are propagated backward along inference trajectories, attributing cumulative and long-horizon effects to intermediate hidden representations. Inspired by (Sundararajan et al., 2017; Dhamdhere et al., 2019), we accumulate policy gradients along the path from a baseline $\mathbf{h}'$ to the original activation $\mathbf{h}$, spanning across activation levels. This path-integral construction yields baseline-aware importance scores that reduce gradient noise, producing robust signals for causal intervention. Crucially, IPG computes these attributions efficiently, requiring only a small number of gradient evaluations without updating model parameters.

To ensure robustness, the IPG attribution score is computed per sample and aggregated across a dataset. Let $\mathcal{D} = \{\mathbf{x}^{(d)}\}_{d=1}^{M}$ be a small supporting dataset of $M$ reasoning questions. For each component index $i$ and each sample index $d$, we compute the per-sample attribution score $\text{IPG}\big(h_i; \mathbf{x}^{(d)}\big)$ (Equation 2). These scores are aggregated into a global importance statistic, for example, the mean

attribution score $S_i$ for component $h_i$, and the top-$p$ components $\mathbf{P}$ are

$$S_i \;=\; \frac{1}{M} \sum_{d=1}^{M} \mathrm{IPG}\big(h_i; \mathbf{x}^{(d)}\big), \quad \mathbf{P} \;=\; \underset{i \in [n]}{\arg \operatorname{top-p}} \; S_i \;, \tag{3}$$

The components with indices within $\mathbf{P} = \{i_1, \ldots, i_p\}$ are treated as the most influential components, for eliciting reasoning behavior that derives the desired outcome. They are further used for subsequent intervention in Section 3.3. Importantly, this identification step is performed *once* for a given frozen model. Once the set $\mathbf{P}$ is identified, interventions can be applied repeatedly without re-computing attributions. This enables direct transfer of reasoning control to other models of the same family (e.g., distilled variants), where we observe consistent effects as demonstrated in Section 4.5. In addition, we provide sensitivity analysis of the number $M$ in Appendix D.1.2. We conclude that even with a relative small $M$, our IPG can accurately select the component that causally drive reasoning ability, which shows the data efficiency of our method.

In our framework, $\mathbf{h}$ can be any interpretable component within the model. Our IPG can be seamlessly integrated with various policy gradient algorithms such as GRPO (Shao et al., 2024). Additionally, it can accommodate any signal that can be expressed as a reward function, such as generation length (see Appendix A.3). To the best of our knowledge, we are the first to integrate gradient-based attribution with policy gradients for interpreting reasoning behavior in LLMs. This approach is beyond purely gradient-based methods by handling sparse and non-differentiable outcome-based signals of reasoning behavior, enabling precise and causal localization of reasoning-critical components. Full illustration of our framework is provided in Figure 1.

### 3.3 CONTROL REASONING BEHAVIOR

To control the reasoning behavior inside the model, we intervene on the components indexed by $i \in \mathbf{P}$ identified in Section 3.2, following interpretability practices (Dai et al., 2022). Formally, for each component $h_i$, we apply a multiplicative scaling factor as

$$h_i = \mathrm{Intervene}(h_i) = \gamma h_i \tag{4}$$

By setting different values of $\gamma$, we can enhance ($\gamma > 1$) or suppress ($0 \le \gamma < 1$) the identified components, thereby controlling the reasoning behavior in the target model (part (c) in Figure 1).

Beyond neuron-level hidden state components $h_i$, we can also intervene on feature-level components by integrating with Sparse Autoencoders (SAEs) (Makhzani & Frey, 2014) for a disentangled feature space. Recent studies (Gao et al., 2025; Galichin et al., 2025) show that sparse features mitigate neuron polysemanticity, enhancing interpretability and steering efficacy. We thus adopt k-Sparse Autoencoders (k-SAEs) (Gao et al., 2025) to transform the original hidden space $\mathbf{h}$ with dimension $n$ into a sparse, higher-dimensional space $\mathbf{f}$ with $m$ larger than $n$, and then decode back to a reconstructed hidden state vector $\hat{\mathbf{h}} \in \mathbb{R}^n$. The encoding and decoding processes are shown as

$$\mathbf{f} = [f_1, f_2, \ldots, f_m] = \mathrm{TopK}(\mathbf{W}_{\mathrm{enc}}(\mathbf{h} - \mathbf{b}_{\mathrm{pre}})) \;, \quad \hat{\mathbf{h}} = \mathbf{W}_{\mathrm{dec}}\mathbf{f} + \mathbf{b}_{\mathrm{pre}} + \epsilon(\mathbf{h}) \;, \tag{5}$$

where $\mathbf{f}$ has non-zero elements $|\mathbf{f}|_0 = k \, (k < m)$, $\mathbf{W}_{\mathrm{enc}} \in \mathbb{R}^{m \times n}$ is the encoder weight, $\mathbf{b}_{\mathrm{pre}} \in \mathbb{R}^n$ is the bias term, and $\mathrm{TopK}(\cdot)$ retains the top $k$ largest values while zeroing others. $\mathbf{W}_{\mathrm{dec}} \in \mathbb{R}^{n \times m}$ is the decoder weight, and $\epsilon(\mathbf{h}) = \mathbf{h} - \hat{\mathbf{h}} \in \mathbb{R}^n$ is the SAE error term. We present results and implementation details in B.1. Empirically (Section 4.2.1), SAE-based interventions yield stronger and more stable control compared to raw neuron scaling, and they enable finer-grained manipulation of reasoning subskills, e.g., numerical calculation and problem decomposition (Section 4.4).

## 4 EXPERIMENT

In this section, we conduct empirical evaluation for our proposed IPG in interpreting and controlling reasoning behaviors in LLMs. We aim to answer the below research questions (**RQs**). **RQ1**: Can we find influential internal reasoning components and control reasoning behavior in language models? **RQ2**: Does IPG find the general and consistent reasoning components across benchmarks? **RQ3**: Can IPG find components that reflect more granular reasoning ability?

Table 1: Comparison of IPG and baseline methods on enhancing and suppressing accuracy across reasoning datasets for Qwen2.5-Math-1.5B-Instruct (Yang et al., 2024) and Llama3.1-8B-Instruct (Grattafiori et al., 2024). Best and second best results are shown in **bold** and underlined format. The arrows ↑ and ↓ means the higher or lower results are better, respectively.

| Method | Enhancing Accuracy ↑ | | | | Suppressing Accuracy ↓ | | | |
|---|---|---|---|---|---|---|---|---|
| | GSM8K | MATH-500 | AIME-2024 | GPQA Diamond | GSM8K | MATH-500 | AIME-2024 | GPQA Diamond |
| **Qwen2.5-Math-1.5B Instruct** | | | | | | | | |
| Original | 82.41 | 63.00 | 10.00 | 24.75 | 82.41 | 63.00 | 10.00 | 24.75 |
| Random | 82.86 | 63.20 | 16.67 | 26.77 | 79.15 | 59.80 | 6.67 | 27.78 |
| Activation | 82.56 | 63.40 | **20.00** | 25.75 | 67.78 | 39.80 | **0.00** | 15.15 |
| R.N. (Rai & Yao, 2024) | 82.26 | 62.20 | 13.33 | 24.24 | 7.13 | 4.40 | **0.00** | 15.15 |
| C.V. (Højer et al., 2025) | 83.85 | 62.60 | 16.67 | 28.78 | - | - | - | - |
| SAE-R (Galichin et al., 2025) | 83.32 | 64.00 | 13.33 | 25.76 | 82.26 | 63.80 | 10.00 | 21.71 |
| **IPG (Ours)** | **84.38** | 64.60 | **20.00** | 30.30 | **0.00** | **1.60** | **0.00** | **0.00** |
| **IPG-SAE (Ours)** | **84.38** | **65.80** | **20.00** | **30.81** | 55.12 | 14.80 | 3.00 | 1.51 |
| **Llama3.1-8B-Instruct** | | | | | | | | |
| Original | 85.89 | 41.80 | 6.67 | 23.23 | 85.89 | 41.80 | 6.67 | 23.23 |
| Random | 86.20 | 41.40 | 6.67 | 27.78 | 84.98 | 39.00 | 10.00 | 26.77 |
| Activation | 85.75 | 43.00 | 13.33 | 25.25 | 81.88 | 28.60 | **0.00** | 14.14 |
| R.N. (Rai & Yao, 2024) | 86.05 | 40.80 | 6.67 | 21.21 | 75.44 | 2.06 | **0.00** | 9.60 |
| C.V. (Højer et al., 2025) | 85.97 | 40.00 | 6.67 | 28.78 | - | - | - | - |
| SAE-R (Galichin et al., 2025) | 86.28 | 41.80 | 10.00 | 26.77 | 85.97 | 43.20 | 10.00 | 22.22 |
| **IPG (Ours)** | **87.41** | 42.40 | 13.33 | 27.27 | **0.00** | **1.40** | **0.00** | **0.00** |
| **IPG-SAE (Ours)** | **87.41** | **44.00** | **20.00** | **29.80** | 85.06 | 33.40 | **0.00** | 1.52 |

## 4.1 EXPERIMENTAL SETUP

**Evaluation Benchmarks.** For comprehensive evaluation, we choose several reasoning-related benchmarks, including GSM8K (Cobbe et al., 2021), Math500 (Hendrycks et al., 2021), AIME2024 (Mathematical Association of America, 2024) and GPQA-Diamond (Rein et al., 2023).

**Target Models.** We experiment with two representative open-source LLMs: Qwen2.5-Math-1.5B-Instruct (Yang et al., 2024) and Llama3.1-8B-Instruct (Grattafiori et al., 2024), and additionally evaluate DeepSeek-Qwen-1.5B (Guo et al., 2025) in Section 4.5. This selection covers different model scales, architectures (Qwen vs. LLaMA), and reasoning paradigms (prompt-elicited vs. training-distilled), highlighting the generalizability of our method.

**Baselines.** We compare our IPG and IPG-SAE (integrated with k-SAE (Gao et al., 2025)) against several baselines: *Random* (RND.), which samples neurons uniformly, and *Activation* (ACT.), which ranks by activation magnitude. We also include *Reasoning Neuron* (R.N.) (Rai & Yao, 2024), an SAE-based method (SAE-R) (Galichin et al., 2025) and *Control Vector* (C.V.) (Højer et al., 2025), which represent interpretable approaches using text patterns and contrastive data pairs for reasoning-related interventions. More details are in Appendix C.2.

**Implementation Details.** We constrain our IPG and baseline methods on the residual stream following (Højer et al., 2025; Galichin et al., 2025) for fair comparison. For the policy gradient algorithm, we employ GRPO (Shao et al., 2024). The external verifier for reasoning outcomes comes from either rule-function based (Appendix A.3), such as generation length (Appendix D.2) or model-based signals for accuracy (using Skywork-Reward-V2-Llama-3.1-8B (Liu et al., 2025)). We integrate k-SAE (Gao et al., 2025) into target models, configured with an expansion factor of 16 and $k = 32$ active features. Following prior work (Cheng et al., 2024; Tang et al., 2025), we consistently employ a greedy decoding strategy during inference with zero-shot setting. For the baseline $h'_i$ mentioned in Equation 2, we choose zero value, i.e. $h'_i = 0$ for all $i \in [n]$, where $n$ is the hidden dimension. We also provide sensitive analysis of choice of baseline in Appendix D.1.4. For intervention, we apply a positive scaling factor $\gamma$ for enhancement and set $\gamma = 0$ for suppression, as described in Section 3.3. More details are provided in Appendix B.

## 4.2 IDENTIFICATION AND CONTROL OF REASONING BEHAVIORS

In this section, we evaluate our IPG and baseline methods both qualitatively and quantitatively on the effectiveness of reasoning ability control across different reasoning-related datasets. (**RQ1**).

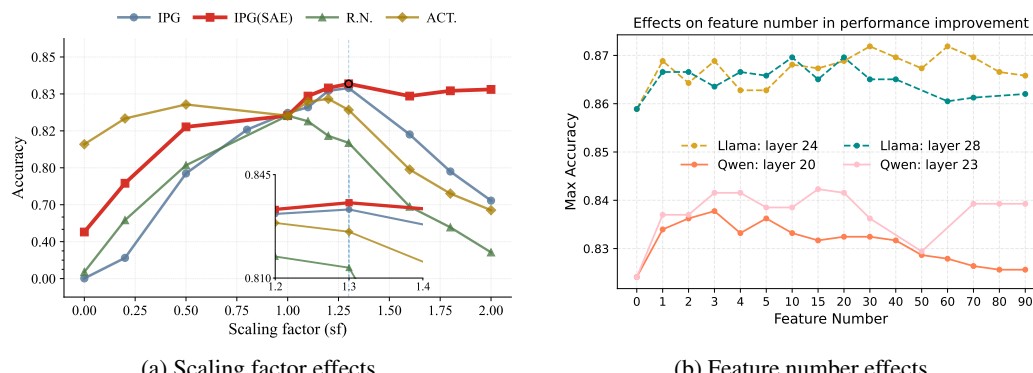

(a) Scaling factor effects.  (b) Feature number effects.

Figure 2: Effects of targeted interventions on reasoning performance. Left: impact of the intervention scaling factor on Qwen2.5-Math-1.5B-Instruct (Yang et al., 2024). Right: impact of the number of selected features on reasoning accuracy.

### 4.2.1 QUANTITATIVE RESULTS ON CONTROLLING REASONING BEHAVIOR

**Analysis 1: Reasoning behavior control with identified components.**   We evaluate our IPG alongside baseline methods for controlling reasoning behavior in two types of language models across multiple reasoning datasets in Table 1. As Control Vector (Højer et al., 2025) is not applicable to the suppression setting, its results are excluded.

**Findings 1: IPG achieves effective control of reasoning behavior.**   In Table 1, IPG achieves excellent and consistent control over the performance of reasoning tasks in both enhancement and suppression settings. Integrating with SAEs further increases our effectiveness. R.N. (Rai & Yao, 2024) and SAE-R (Galichin et al., 2025) provide only modest improvements, while C.V. (Højer et al., 2025) shows mixed gains and fails on MATH500 (Hendrycks et al., 2021). These results highlight the limitations of approaches relying on text patterns or input contrastive pairs, which cannot reliably uncover components that causally drive reasoning. Overall, our findings suggest that causality-aware, outcome-driven attribution, especially when integrated with disentangled feature representations, offers a more faithful and robust mechanism for steering LLM reasoning than prior baselines. This shows the advantages over existing methods paradigms as shown in Figure 1.

**Analysis 2: Fine-grained control of reasoning behavior.** We vary the scaling factor $\gamma$ and the number of intervened components (top-$p$ SAE features) to probe how sensitively reasoning performance responds (see Figure 2). We randomly select 2 layers within the two models (Figure 2b) and choose the top-ranked features based on our attribution score in Equation 3.

**Finding 2: IPG enables precise modulation of reasoning behavior.** As shown in Figure 2a, baseline interventions yield only minor or unstable effects, whereas IPG produces clear, predictable changes that amplifying identified components consistently improves reasoning, while suppressing them degrades it. The effect is smoothly controlled by the scaling factor. Notably, IPG with SAE remains stable even as baselines collapse. In figure 2b, we can observe that reasoning-critical features are concentrated in the top-ranked subset, with even single-feature interventions yielding substantial gains. Overall, IPG achieves more effective and tunable control of reasoning than prior methods.

### 4.2.2 QUALITATIVE RESULTS ON CONTROLLING REASONING BEHAVIOR

**Analysis 3: Response difference before and after intervention.**   We showcase an example from GSM8K (Cobbe et al., 2021) before and after applying intervention (see in Section 3.3) in the neurons identified by our IPG, demonstrating the impact on improving and suppressing reasoning ability, as illustrated in Figure 3. More examples are provided in Appendix D.4.2.

**Finding 3: IPG precisely identifies reasoning-critical components.** As shown in Figure 3, the original response fails to follow the strict requirement. The contrast shows that interventions on IPG-identified components have causal effects on intermediate reasoning steps. For the suppressing setting, the model can not even get the equation correct. This demonstrates that IPG does not merely push the model toward a correct final output, it also improves the internal reasoning trajectory,

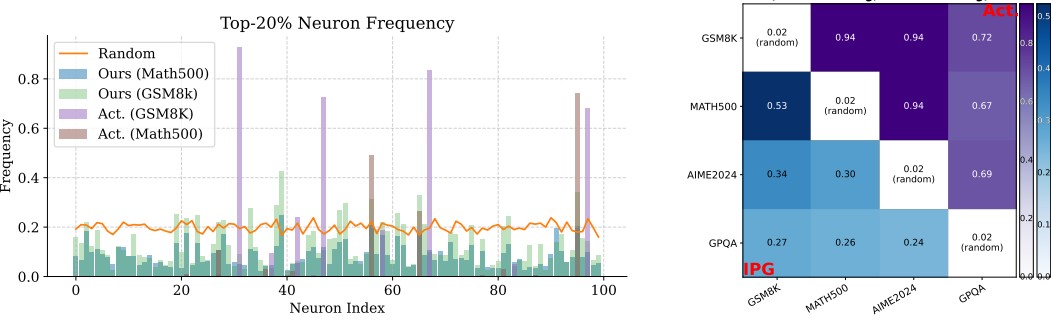

> **Question**
> Carlos is planting a lemon tree. The tree will cost $90 to plant. Each year it will grow 7 lemons, which he can sell for $1.5 each. It costs $3 a year to water and feed the tree. How many years will it take before he starts earning money on the lemon tree?

**Original**

To determine how many years it will take for Carlos to start earning money…
6. **Solving for n:**
$$n \geq \frac{90}{7.50} = 12$$ ❌
Therefore, it will take Carlos \\(\\boxed{12}\\) years to start earning money on the lemon tree.

**Enhanced Reasoning**

To determine how many years it will take for Carlos to start earning money …
6. **Finding the break-even point:**
We need to find the smallest n such that the total profit is positive.
$$7.50 \times n - 90 > 0$$
Solving for $n$ : $n > \frac{90}{7.50} = 12$ ✅
Since $n$ must be a whole number, we round up to the next whole number. Therefore, $n = 13$.
Thus, it will take \\(\\boxed{13}\\) years before Carlos starts earning money on the lemon tree.

**Suppressed Reasoning**

To determine how many years it will take for Carlos to earn money…
Setting the total revenue equal to the total costs to find the break-even point:
$$7.5n = 90 + 3n$$
7. **Solve for n:**
$$4.5n = 90$$
$$n = \frac{90}{4.5} = 20$$ ❌ ❌
Therefore, it will take \\(\\boxed{20}\\) years for Carlos to start earning money from the lemon tree.

Figure 3: Responses generated by Qwen2.5-Math-1.5B-Instruct (Yang et al., 2024) on one example in GSM8K (Cobbe et al., 2021), including both original and intervened outputs, with the model's reasoning ability elicited to arrive at the correct answer.

making the reasoning process more complete and coherent. Motivated by this finding, we provide further exploration on the granular aspects of reasoning behaviors, as presented in Section 4.4.

We also conduct another experiment on reasoning mechanism interpretability, i.e., reasoning length related mechanism analysis (Figure 5). We examine the controllability of trajectory length as a concrete example of reasoning mechanism interpretability, where interventions successfully modulate the output length of DeepSeek-Qwen-1.5B (Guo et al., 2025) without sacrificing final answer accuracy. This demonstrates that IPG can identify neurons corresponding to different reasoning abilities, with interventions leading to interpretable effects across multiple reasoning facets, highlighting the generality of our approach.

(a) Frequency of top $p = 20$ neuron indices for GSM8K (Cobbe et al., 2021) and MATH500 (Hendrycks et al., 2021).

(b) Jaccard similarity between neuron indices across datasets.

Figure 4: Left: histogram of neuron index frequencies, highlighting the top-$p = 20\%$ neurons identified by our method on GSM8K (Cobbe et al., 2021) and MATH500 (Hendrycks et al., 2021). Right: similarity matrix comparing neuron indices discovered in 4 different datasets, highlighting shared reasoning structure across tasks, with diagonal showing random baseline.

### 4.3 GENERAL REASONING MECHANISM ACROSS TASKS

In this part, we show whether the components identified by our IPG and the baseline method can transfer seamlessly from one dataset to others, finding the general and consistent components that truly causally drive the reasoning behavior (**RQ2**).

**Analysis 4: Components consistency, similarity and performance across dataset.** To investigate how consistent the reasoning components are , we investigate the transferability of these components by applying those derived from GSM8K (Cobbe et al., 2021) to MATH500 (Hendrycks et al., 2021), and vice versa. The resulting accuracy is presented in Table 2. Furthermore, we visualize the frequency of reasoning components selected across samples in Figure 4a. For each sample, we compute an importance score per component (via IPG attribution from Sec. 3.2 or raw activations) and select the top-$p$ neurons. Aggregating these selections across all samples yields a per-neuron frequency distribution. High overlap in these frequency distributions indicates that the neurons exhibit consistent behavior. We further quantify cross-dataset consistency using Jaccard similarity, which is

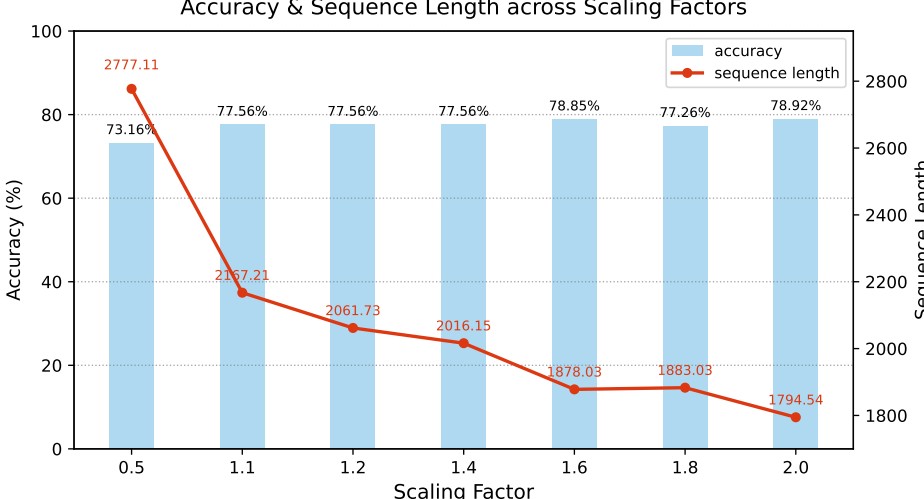

Figure 5: Length control in GSM8K (Cobbe et al., 2021) for DeepSeek-R1-Distilled-Qwen-1.5B (Guo et al., 2025). IPG precisely identifies neurons that is related with reasoning length, suggested by effective control while maintaining high accuracy.

defined as $\frac{|\mathbf{P_A} \cap \mathbf{P_B}|}{|\mathbf{P_A}| + |\mathbf{P_B}| - |\mathbf{P_A} \cap \mathbf{P_B}|}$ in Figure 4b, where $\mathbf{P_A}$ and $\mathbf{P_B}$ are neuron indices identified in the two datasets. Diagonal elements represent the RND. baseline, upper triangular elements show the Act baseline, and lower triangular elements depict our proposed IPG.

**Finding 4: IPG finds consistent reasoning mechanisms across diverse tasks.** Cross-dataset transfer results as shown in Table 2, suggests that IPG achieves the most accuracy gains, showing our consistence across tasks. By comparing with existing interpretability baseline methods (details in Section 4.1), we conclude that our IPG accurately identified the inherent reasoning components, demonstrating stronger cross-benchmark generalization. Figure 4a shows that IPG identifies components

Table 2: Performance on MATH500 (Hendrycks et al., 2021) and GSM8K (Cobbe et al., 2021), for cross-dataset transfer. Best and second best results are shown in **bold** and underlined format.

| Method | MATH-500 → GSM8K | | GSM8K → MATH-500 | |
|---|---|---|---|---|
| | Acc. (%) ↑ | Avg. Tok. | Acc. (%) ↑ | Avg. Tok. |
| Original | 82.41 | 319.68 | 63.00 | 569.35 |
| Activation | 82.94 | 317.17 | 63.60 | 567.39 |
| R.N. (Rai & Yao, 2024) | 82.87 | 312.63 | 63.20 | 556.97 |
| C.V. (Højer et al., 2025) | 83.39 | 320.03 | 63.60 | 568.55 |
| **IPG (Ours)** | 83.47 | 320.06 | 64.00 | 564.59 |
| **IPG-SAE (Ours)** | **84.08** | 315.86 | **64.20** | 558.94 |

with consistent cross-benchmark overlap, revealing reasoning-relevant features shared between datasets. In contrast, the Activation baseline tends to pick neurons with persistently large magnitudes, which does not reliably control reasoning performance, suggested by the inferior and unstable accuracy in Table 1. As quantified in Figure 4b: IPG shows moderate, not extreme overlap across selected sets, whereas Activation's high similarity reflects only magnitude-driven bias. Given that these reasoning benchmarks share underlying skills but diverge considerably in difficulty, the moderate Jaccard similarity is expected. This suggests that we successfully capture general, transferable reasoning components while retaining the benchmark-specific adaptability.

## 4.4 FINE-GRAINED REASONING BEHAVIOR DISCOVERY

We further look into the identified reasoning components to reveal its granular aspects on reasoning behaviors, shedding some lights on the inner workings of LLMs on reasoning ability (**RQ3**).

**Analysis 5: Fine-grained aspects of reasoning behavior discovery.** We employ causal interventions to analyze reasoning behavior at a fine-grained level. Specifically, we intervene on individual components, e.g., single neuron or feature ranked by IPG scores (Eq. 3), and assess their impact on reasoning behavior over GSM8K (Cobbe et al., 2021). Each component is evaluated along four reasoning dimensions (*Semantic*, *Decomposition*, *Thoroughness*, and *Calculation*). GPT-5-mini is used to summarize and label the model's reasoning behaviors, and representative examples of enhancing a single neuron versus an SAE feature are shown in Figure 6. Additional experimental details are in Appendix B.2 with Table 10 illustrate these reasoning dimensions.

**Findings 5: IPG shows granular aspects on reasoning behaviors.** As shown in Figure 6, enhancing neuron #940 alters both *Thoroughness* and *Semantic*, whereas boosting SAE feature #8053 primarily improves *Calculation*. Consistent with prior observations of neuron polysemanticity (Mu & Andreas, 2020; Olah et al., 2020), intervening on a single neuron frequently impacts multiple dimensions simultaneously. In contrast, SAE-derived features are more disentangled as interventions on an individual feature tend to affect a single reasoning aspect. These causal interventions demonstrate that neurons and sparse features play distinct functional roles in multi-step reasoning, and that disentangled feature spaces enable more targeted control. Additional case studies are provided in Appendix D.4.1.

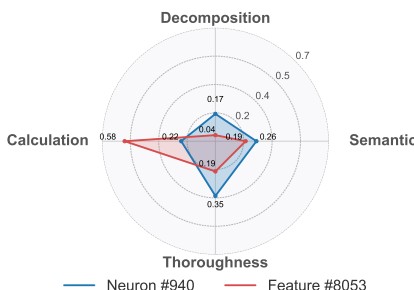

Figure 6: Dual radar plots of neuron- and feature-level interventions: top for enhancing neuron #940 and feature #8053.

### 4.5 ROBUSTNESS OF FOUND MECHANISMS UNDER DISTILLED MODELS

**Analysis 6: Transferring IPG-identified neurons into reasoning-distilled models.** In addition to general purpose models with prompt-elicited reasoning, we probe the robustness of IPG-identified mechanisms under a reasoning-distilled model. Specifically, we intervene on neurons identified in Qwen2.5-Math-1.5B-Instruct (Yang et al., 2024) within DeepSeek-R1-Distilled-Qwen-1.5B (Guo et al., 2025).

**Findings 6: Components identified by IPG can be seamlessly transferred.** As shown in Table 3, steering the inherited neurons in the DeepSeek-distilled model consistently improves performance across datasets. This indicates that the core reasoning neurons remain crucial even when the underlying model is dis-

| Model | GSM8K | Math500 | AIME2024 |
|---|---|---|---|
| DeepSeek-R1-Distill-Qwen-1.5B | 75.21 | 61.40 | 16.67 |
| + IPG Neuron Enhancement | **77.26** | **62.00** | **26.67** |

Table 3: Results of transferred IPG neuron enhancement with reason-distilled models.

tilled for reasoning. This result not only validates the ability of IPG to pinpoint fundamental, robust neurons for reasoning, but also offers an insight for the mechanism of model distillation, suggesting that distillation might strengthen the key reasoning-related components that are already present in the base model, rather than reshaping the internal reasoning structure.

## 5 CONCLUSION

In this work, we propose IPG, a novel framework for interpreting LLM reasoning behavior that based on causality-aware and outcome-oriented principles. Our IPG applies gradient-based methods to identify influential internal components that causally contribute to reasoning behavior. By incorporating policy gradients (Schulman et al., 2017), we attribute the outcomes that depend on cumulative and long-range effects in reasoning behavior and address the challenge of sparse and non-differentiable outcome signals. Empirically, IPG provides effective and interpretable identification and control of reasoning behavior across different types of LLMs. Additionally, IPG exhibit excellent transferability both in reasoning datasets and between prompt-elicited and training-induced model variants. Our IPG offers interpretable solution to control reasoning ability in LLMs, though challenges remain such as more effective intervention. Future work includes extending our IPG framework to interpret domains where performance is hard to quantify, such as the emotional intelligence and creativity in LLMs or alternative model structures (e.g., LlaDA (Nie et al., 2025)), and to precisely manipulate model behavior by steering the identified components.

## 6 REPRODUCIBILITY STATEMENT

We have made great efforts to ensure the reproducibility of our results. The experiment setup, including model configurations, training steps and evaluation datasets, metrics, is clearly described in the main paper and appendix. We believe this will help other researchers reproduce our work and further advance the field. The reproducible code will be released upon the acceptance of this paper.

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

# A  ALGORITHMS OF IPG

In this section, we provide two main algorithms of our method in order to clearly illustrate the key steps of IPG. The first algorithm focuses on *identifying reasoning-critical components* by attributing outcome-weighted signals to internal representations. The second algorithm is based on these identified components to *control reasoning behavior* of the model. (see more details in 3.2,3.3)

## A.1  IDENTIFYING REASONING COMPONENTS

We introduce how to identify reasoning-critical components. We use a support set of $\mathbb{N}$ samples, which can be either a subset or the full dataset of any reasoning-related benchmarks like GSM8K (Cobbe et al., 2021). We denote output residual stream of transformer block as $\mathcal{F}(\cdot)$ which takes a sample as input and we can get the hidden state of residual stream. Here, $\Phi$ can be either the identity map, corresponding to neuron-level IPG, or the SAE encoder, corresponding to feature-level IPG. After that, we can localize the reasoning-critical components using IPG. For the calculation of attribution, we use Riemann approximation to estimate integral:

$$IPG(h_i; \mathbf{x}) = (h_i - h_i') \cdot \frac{1}{q} \sum_{k=1}^{q} \mathbb{E}_{\tau \sim \pi_\theta} \left[ \sum_{t=0}^{T} \frac{\partial}{\partial h_i} \log \pi_\theta \left( a_t \mid s_t; h' + \frac{k}{q}(h - h') \right) \cdot A^\pi(s_t, a_t) \right]$$

(6)

where $\mathbf{h}' = [h_1', \ldots h_m']$ is a relative baseline of $\mathbf{h}$ and $q$ is the number of discrete steps or partitions used in the Riemann approximation to estimate the integral. $\mathbf{x}$ is the data sampel. Notably, we use $\mathbf{u}$ in the following Alg. 1 and 2 to denote the representation of either neuron or SAE feature. More details about $F$ are provided in the Appendix A.3.

---

**Algorithm 1:** IPG: Identifying Reasoning Components

---

**Input:** A support set of samples $X = \{\mathbf{x}_1, \ldots, \mathbf{x}_N\}$, $p \in \mathbb{N}$

1 **for** $j = 1$ **to** $N$ **do**
2     Extract $\mathbf{h} = [h_1, h_2, \ldots, h_n]$ from $\mathcal{F}(\mathbf{x}_j)$
3     $\mathbf{u} = [u_1, u_2, \ldots, u_m] = \Phi(\mathbf{h})$
4     **for** $i = 1$ **to** $m$ **do**
5         $S(u_i; X)$ += $IPG(u_i; \mathbf{x})$ ;         // Eq. 2
6 $\mathbf{P} = \{i_1, \ldots, i_p\} = \arg\text{top-p}_{i \in [n]} \{S(u_i; X) \mid u_i \in \mathbf{u}\}$ ;    // Eq. 3
**Output:** $\mathbf{P}$

---

**Algorithm 2:** IPG: Controlling Reasoning Behavior

---

**Input:** A set of component indexes $P = \{i_1, \ldots, i_p\}$, hidden state extracted from residual stream $\mathbf{h} = \{h_1, h_2, \ldots, h_n\} \in \mathbb{R}^n$, scaling factor $\gamma \in \mathbb{R}$

1 $\mathbf{u} \leftarrow [u_1, u_2, \ldots, u_m] = \Phi(\mathbf{h})$ ;         // Eq. 5
2 Create $\mathbf{u}' \leftarrow \{u_1', u_2', \ldots, u_m'\} = \mathbf{p}$
3 **for** $k \leftarrow 1$ **to** $\tau$ **do**
4     $u_{i_k}' \leftarrow \gamma u_{i_k}'$ ;         // Scaling, Eq. 4
5 **if** $\Phi = $ Identity **then**
6     $\mathbf{h} \leftarrow \mathbf{u}$
7 **else if** $\Phi = $ SAE-encoder **then**
8     $\epsilon \leftarrow \mathbf{h} - (\mathbf{W}_{\text{dec}}\mathbf{u} + \mathbf{b}_{\text{dec}})$ ;
9     $\mathbf{h} \leftarrow \mathbf{W}_{\text{dec}}\mathbf{u}' + \mathbf{b}_{\text{dec}} + \epsilon$ ;         // Eq. 5
**Output:** $\mathbf{h}$

---

## A.2  CONTROLLING REASONING BEHAVIOR

After identifying reasoning-critical components, we further demonstrate how to directly control the reasoning behavior of the model. Given a set of target component indexes $P$, we first encode the

hidden state $\mathbf{h}$ into the representation space $\mathbf{s}$ using $\Phi$, which can be either the identity mapping (for neuron-level control) or the SAE encoder (for feature-level control). We then intervene on the selected components by scaling their activations with a factor $\gamma$, yielding a modified representation $\mathbf{s}'$. Finally, we decode $\mathbf{s}'$ back to the residual stream. Notably, in the SAE setting, the reconstruction also retains the error term to ensure faithful recovery of the original hidden state structure (see more details in B.1 about this).

## A.3 VARIANTS OF IPG

In this section, we provide variants of our IPG that supports different algorithms of policy gradient (Mnih et al., 2016), as mention in Section 3.2.

In GRPO (Shao et al., 2024), the structure of IPG remains identical, except the advantage $A_t$ is replaced with a group-relative advantage:

$$A_t = \frac{r_t - \text{mean}(\{r_1, r_2, \ldots, r_G\})}{\text{std}(\{r_1, r_2, \ldots, r_G\})}. \tag{7}$$

where $T$ is the sequence length, $N$ is the sample number, and $\pi_\theta$ is the policy parameterized by $\theta$, i.e., the LLM to be interpreted. $r_t$ is the reward at step $t$. $s_t^{(k)}$ is the prefix of the sequence $k$ until $t$ steps, $a_t^{(k)}$ is the next token in the sequence $k$, and $A^\pi(s_t, a_t)$ is the advantage function that estimates benefit of selecting $a_t$ over the baseline, often derived from rewards like reasoning accuracy scores (Zhong et al., 2025).

$$\text{IPG}^{\text{GRPO}}(h_i; \mathbf{x}) = (h_i - h_i') \int_0^1 \frac{1}{N} \sum_{\tau=1}^{N} \sum_{t=1}^{T} \frac{\partial}{\partial h_i} \log \pi_\theta(a_t \mid s_t; h' + \alpha(h - h')) A_t^{\text{GRPO}} d\alpha, \tag{8}$$

Here, $A_t$ denotes the advantage function, which measures how much better the chosen action $a_t$ is compared to the expected baseline at state $s_t$. $\mathbf{x}$ is the data sample. As we can see in Equation 8, our IPG can be seamlessly integrated with diverse policy gradient algorithms by integrating their advantage calculation.

For the reward source, we support both **rule-based** and **model-based** signals. In the rule-based setting, the reward $r$ is defined as a binary indicator of final-answer correctness:

$$r = \mathbb{I}\{\hat{y} = y\} = \begin{cases} 1, & \text{if } \hat{y} = y, \\ 0, & \text{otherwise}. \end{cases} \tag{9}$$

In contrast, model-based rewards provide a continuous evaluation of reasoning quality. For example, given a scoring model $M(\cdot)$, we can assign rewards as

$$r = M(x, \hat{y}), \tag{10}$$

where $M$ may return partial credit (e.g., proportion of correct intermediate steps) or preference scores from human-aligned models. This yields a richer supervision signal than binary correctness, enabling finer-grained attribution of reasoning ability. A comparison of these reward sources is presented in Appendix D.1.3.

## B IMPLEMENTATION DETAILS

The reproducible code will be released upon the acceptance of this paper.

### B.1 SAE TRAINING

**Dataset for training k-SAEs:** In this section, we provide details about training the k-SAE. Followed by Galichin et al. (2025), We train SAE on the activations of the model using the full *OPENTHOUGHT-114K* dataset (OpenThoughts, 2025), which is composed of high quality reasoning trace generated by DEEPSEEK-R1 including math, science, code, etc. To elicit consistent chain-of-thought style activations when harvesting activations, we apply the following chat template:

```
<system>Please reason step by step and put your answer within \boxed{}.

</system><|user|>{question}<|assistant|>{deepseek_reasoning}
{deepseek_solution}
```

**Training K-SAE on Hidden Space:** As shown in equation 5, the k-SAE contains an encoder $\mathbf{W}_{\text{enc}}$ and a decoder $\mathbf{W}_{\text{dec}}$ initialized with the same parameters. The reconstruction loss we use is a standard mean squared error (MSE) loss defined as

$$\mathcal{L}(\mathbf{h}) = \|\mathbf{h} - \hat{\mathbf{h}}\|_2^2 \tag{11}$$

where $\mathbf{h}$ is the original hidden state vector and $\hat{\mathbf{h}}$ denotes its reconstruction (excluding the residual error term). The dimension of the sparse representation space is 24576 for Qwen2.5-Math-1.5B-Instruct (Yang et al., 2024) and 65536 for Llama-3.1-8B-Instruct (Grattafiori et al., 2024)corresponding to an expansion factor of 16. We train k-SAE on a single NVIDIA H20 GPU, taking approximately 7 hours for Qwen and 30 hours for Llama.

**Training Results:** We present the details about the training of k-SAEs, including hyper-parameters and learning curves. We use the learning rate of 0.005 with batch sizes of 32, context length of 2048 tokens. For the training curve, we leverage Fraction of Variance Unexplained (FVU)(Makhzani & Frey, 2014), which is a related metric of interest, measuring the total amount of the original activation that is not "explained" or reconstructed well by k-SAE. FVU is formally defined as

$$FVU = \frac{\mathcal{L}(\mathbf{h})}{\text{var}[\mathbf{h}]} \tag{12}$$

where $\mathbf{h}$ is the hidden state, $\mathcal{L}(\mathbf{h})$ is defined in Eq. 11 and var represents the varience of $\mathbf{h}$. A lower FVU indicates better reconstruction performance since more original activation is captured by the k-SAE model.The training curves in Fig. 7 show that the k-SAE perform well on the FVU metric.

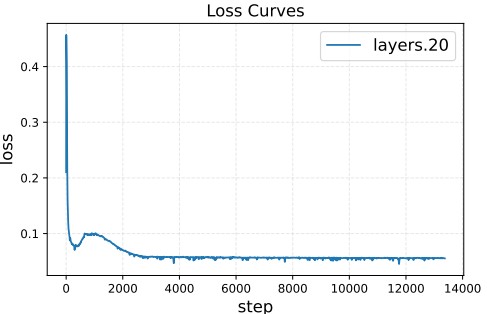 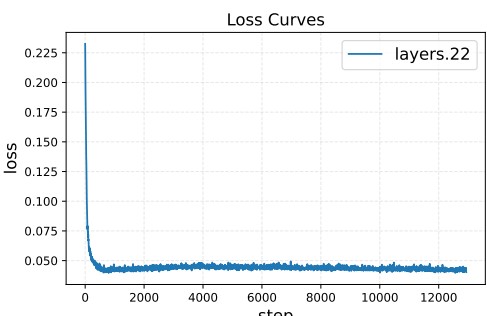

(a) Training Curve for Qwen2.5-Math-1.5B-Instruct    (b) Training curve for Llama-3.1-8B-Instruct

Figure 7: SAE Training Curve

**Discussion: The Role of SAE Error Node** While the k-SAE can effectively reconstruct the hidden state $\hat{\mathbf{h}}$, residual gaps inevitably remain. Recent studies (Lindsey et al., 2025; Marks et al., 2025) highlight that explicitly modeling error nodes is essential to fill these gaps, as it gives a principled decomposition of model behaviors into contributions from interpretable features and error components not yet captured by our SAEs. In our setting, we retain the SAE error term to preserve the reliability of our downstream intervention, thus ensuring that our modification is incremental without degrading the model's performance.

### B.2    FINE-GRAINED REASONING MECHANISM DISCOVERY

In this section, we provide more implementation details on fine-grained reasoning mechanism discovery. Specifically, we rank reasoning components by their IPG scores and automatically label the top 30 neurons and SAE features identified from Qwen2.5-Math-1.5B-Instruct using a GPT-5-mini API respectively. For interpretation, we manually decompose the comprehensive reasoning into

four hierarchical layers in reasoning: *Semantic Comprehension*, *Problem Decomposition*, *Reasoning Depth* and *Numerical Accuracy*. Importantly, our analysis is causal: we directly intervene on each neuron or feature, collect all newly-correct answers after steering, and compare pre- and post-intervention responses. The resulting behavioral changes are then judged by the API to correspond to one of the four reasoning abilities. These categories align with the labeling scheme defined in the system prompt used for API calls, which is shown below.

```
You are a careful evaluator. Output JSON only (an object). Do not include
any extra text.
Allowed labels (with concise role notes):
1. FineGrainedSemanticComprehension [Semantic Layer]
Evaluates precise understanding of wording, references, modifiers, units.
Focus: resolving ambiguity/negation/quantifiers so the problem is
    interpreted correctly.
2. ProblemDecompositionAndLogicalSequencing [Planning Layer]
Evaluates whether the task is decomposed into necessary subgoals and
    ordered coherently. Focus: a plan that covers all required steps and
    aligns with the objective.
3. ReasoningDepthAndThoroughness [Logical Execution Layer]
Evaluates completeness and correct application of rules/conditions
during non-numerical reasoning.
Focus: ensuring all relevant conditions are correctly applied during
    each step, variable states are continuously tracked, and implicit
    logical premises are not omitted.
4. NumericalAccuracyAndCalculation [Calculation Layer]
Evaluates correctness of arithmetic/algebra/probability and unit
    conversions.
Focus: operation order, numeric consistency, and appropriate rounding.
```

## C    EXPERIMENT SETTINGS

In this section, we provide details on the experiment settings (Section 4.1), including our evaluation pipeline and the baseline implementation.

### C.1    EVALUATION SETTINGS

For all benchmarks, we employ the lm-evaluation-harness framework (Gao et al., 2024) as the evaluation tool. For Qwen2.5-Math-1.5B-Instruct, we set the maximum tokens to 2048, while for Llama-3.1-8B-Instruct, we set the maximum new tokens to 8192. Followed by prior work(Dutta et al., 2024) , Interventions with IPG are applied to the mid-to-late transformer blocks, specifically layers 20–26 for Qwen and layers 22–28 for Llama. To avoid randomness and ensure reproducibility, we set do sample to false, which is equivalent to greedy decoding strategy (temperature=0). Accordingly, we employ Accuracy@1 as our metric.

### C.2    BASELINES

**Random.**    For comparison, we construct a random baseline by uniformly and randomly selecting neurons within the hidden states **h** and applying interventions based on these neurons. The final results of this baseline are averaged across multiple random seeds.

**Activation.**    Internal neuron activations contain meaningful information. Additionally, Dai et al. (2022) demonstrates the ability to edit internal neurons to influence neural network performance. Building on this, we adopt a similar approach, performing intervention directly on the original activations in the hidden state. For a fair comparison, we constrain our intervention to the residual stream of a language model.

**Reasoning Neuron**    , as described in Rai & Yao (2024), the focus is on analyzing activation patterns, such as average activation values in the feed-forward (FF) layer, to identify reasoning neurons

and interpret them using predefined concepts $C$, such as arithmetic operations $C_{add}$ with key tokens like "add", "+". We extend this approach to operate on the residual stream of transformer blocks (Vaswani et al., 2017). By leveraging all provided concepts, we identify related neurons and use them for intervention.

**Control Vector**, introduced by Højer et al. (2025), employs representation engineering by extracting model activations from the residual stream of a large language model (LLM) during a reasoning task. These activations are used to derive a control vector for inference-time intervention. The control vector is constructed from positive and negative pairs: positive pairs consist of Chain-of-Thought (CoT) prompts where the model produces correct outputs, while negative pairs comprise 75 random character strings sampled from the alphabet (A–Z). Then PCA is applied and the first principal component in the control direction is treated as the most reasoning-related direction. Following the open-source repository, we use 30% of the samples from each dataset to compute the control vector.

**SAE-Reasoning**, as introduced by Galichin et al. (2025), employ Sparse Autoencoders (SAEs), learn a sparse decomposition of latent representations of a neural network into interpretable features, to identify features that drive reasoning in the DeepSeek-R1 (Guo et al., 2025) series of models. They extract candidate "reasoning feature" from SAE representations, that based on specific tokens such as {"wait", "but"}, associating feature activation with these tokens. For each feature, *reason-score* is computed based on its feature activation on the *OPENTHOUGHT-114K* (OpenThoughts, 2025). After that, following the releasing implementation, we intervene the feature with top *reason-score*.

**STA.**, introduced by (Wang et al., 2025a), this approach leverages SAE-decoupled representations to identify and manipulate target atoms, isolating and modifying disentangled knowledge components to enhance the behavior of large language models (LLMs). To obtain the vector for controlling length, we follow their setup and use only one sample: "1+1=?", where the positive prompt is a direct answer "2" and the negative prompt is a redundant, complex answer. We apply the derived vector on the residual stream.

# D  MORE RESULTS

In this part, we present more results of our experiments part and studies for better illustrations of our IPG.

## D.1  SENSITIVE STUDY

### D.1.1  K-SAE HYPERPARAMETER

We adjust different $k$ values to evaluate the effects of the k-SAEs (Gao et al., 2025) on reasoning behavior control as shown in Table 4. The results are based on Qwen2.5-Math-1.5B-Instruct (Yang et al., 2024) layer 20.

| $k$ | GSM8K | |
| --- | --- | --- |
| | **Acc. (Enhance)** $\uparrow$ | **Acc. (Suppress)** $\downarrow$ |
| Original | 82.41 | 82.41 |
| 8 | **84.38** | 82.26 |
| **32** | **84.38** | **55.12** |
| 64 | 83.70 | 81.96 |

Table 4: Investigation of the impact of varying TopK parameters in k-SAE Belitsky et al. (2025) on controlling reasoning behavior in Qwen2.5-Math-1.5B-Instruct (Yang et al., 2024) for GSM8K (Cobbe et al., 2021). TopK refers to preserving the top $k$ features while deactivating others in the feature space, as described in Section 3.3.

In Table 4, as the TopK parameter $k$ increases, the ability to control reasoning behavior, including both enhancing and suppressing, either persists or begins to diminish. We demonstrate that our

results are not heavily dependent on the choice of Sparse Autoencoders (SAEs). Instead, our focus is on the accurate identification and intervention of reasoning features. Consequently, we select $k = 32$ for the k-SAE (Gao et al., 2025) across all experiments.

### D.1.2 IPG ATTRIBUTION HYPERPARAMETER

We conduct experiment on the effects of different number $p$ of components identified, as stated in Section 3.2. In Table 5, as the number of top-$p$ neurons increases, the control over reasoning

| | GSM8K | |
|---|---|---|
| $p$ | Acc. (Enhance) ↑ | Acc. (Suppress) ↓ |
| Original | 82.41 | 82.41 |
| 10 | 83.55 | 28.13 |
| 20 | 83.70 | 25.55 |
| 30 | **83.78** | 17.82 |
| 40 | **83.78** | 14.78 |
| 50 | 83.17 | **10.69** |

Table 5: Evaluation on the impact of varying the top-$p$ number of neurons in 3.3 in Qwen2.5-Math-1.5B-Instruct (Yang et al., 2024) for GSM8K (Cobbe et al., 2021).This evaluation is based on our k-SAE implementation (Gao et al., 2025) with a fixed $k = 32$.

behavior becomes more effective. However, beyond a certain point, adding more neurons does not yield further improvements. This is because reasoning-related concepts may be distributed across multiple neurons, and the top neurons already exert significant control, while additional neurons lack substantial reasoning-related functionality.

We also conduct sensitivity analysis on our choice of batch size $M$. We perform an ablation on the number of samples used to compute the IPG directions, varying it from 1 to 500. All other settings follow the main experimental protocol (Table 6): Qwen2.5-Math-1.5B-Instruct (Qwen Team, 2024) evaluated on GSM8K (Cobbe et al., 2021) under greedy decoding. "Enhance" reports accuracy after strengthening the identified reasoning components (higher is better), while "Suppress" reports accuracy after weakening them (lower is better). The original accuracy of the model is 82.41%.

Results are presented in Table 6. When $M$ is very small ($\leq 10$), the gradient estimates exhibit high variance, which degrades component identification and leads to weak performance. As it increases to $\geq 20$, variance decreases rapidly, yielding smoother and more reliable gradient signals. This stabilizes both enhancement and suppression, with suppression improving dramatically.

### D.1.3 REWARD SIGNALS

In Table 7, we report the performance of IPG in improving accuracy across reasoning datasets for Qwen2.5-Math-1.5B-Instruct (Yang et al., 2024) using different reward signals, including rule-based and reward model-based signals, as detailed in Appendix A.3. The scaling intervention (see Section 3.3) is applied at the 24th layer. As shown in Table 7, reward model-based signals outperform rule-based signals in most datasets, suggesting that reward models capture richer information

| | GSM8K | |
|---|---|---|
| # Samples | Acc. (Enhance) ↑ | Acc. (Suppress) ↓ |
| 1 | 82.79 | 82.03 |
| 10 | 83.17 | 79.37 |
| 20 | 83.17 | 06.52 |
| 50 | 83.54 | 12.81 |
| 100 | 83.39 | 12.81 |
| 500 | **83.55** | **12.05** |

Table 6: Effect of batch size (# samples used to compute IPG directions) on steering performance. Model: Qwen2.5-Math-1.5B-Instruct on GSM8K (Cobbe et al., 2021). Original accuracy: 82.41%.

| Model | Reward Method | GSM8K | MATH-500 | AIME-2024 | GPQA Diamond |
|-------|---------------|-------|----------|-----------|--------------|
| Qwen2.5-Math-1.5B | Rule Fuction | 84.00 | 64.20 | 20.00 | 26.77 |
| | Reward Model | 84.38 | 64.40 | 16.67 | 30.30 |

Table 7: Performance of IPG in enhancing accuracy across reasoning datasets for Qwen2.5-Math-1.5B-Instruct (Yang et al., 2024) under different reward signals.

| $h'$ | GSM8K | |
|------|-------|---|
| | Acc. (Enhance) $\uparrow$ | Acc. (Suppress) $\downarrow$ |
| Zero vector | 84.38 | **0.00** |
| Mean hidden state | **84.45** | 21.15 |
| Random noise | 84.00 | 78.16 |

Table 8: Sensitivity analysis on the choice of baseline representation $h'$. The model is Qwen-Math-2.5-1.5B-Instruct (Qwen Team, 2024) evaluated on GSM8K (Cobbe et al., 2021). Original accuracy before any intervention: 82.41%.

or better represent reasoning outcomes, thereby enhancing IPG's effectiveness in identifying and controlling reasoning behavior.

### D.1.4 BASELINE CHOICE.

To further verify the robustness of IPG, we conduct additional experiments on different choices of the baseline hidden state $h'$ using Qwen-Math-2.5-1.5B-Instruct (Qwen Team, 2024) on the GSM8K (Cobbe et al., 2021) dataset. We report accuracy under greedy decoding as described in Section 4.1. The original accuracy of the model is 82.41%. We compare three baseline strategies: (i) the zero vector, (ii) the mean hidden state computed over the training set, and (iii) random Gaussian noise $\mathcal{N}(0, I)$ as an explicit non-informative reference.

Results are shown in Table 8. Using the zero vector or the mean hidden state yields nearly identical enhancement performance (84.38% and 84.45%, respectively) and both enable almost perfect suppression (0.0% and 21.15%). In contrast, random noise performs considerably worse on suppression (78.16%), confirming that it does not constitute a meaningful baseline representation. These findings demonstrate that IPG is highly robust to the specific choice of $h'$ as long as it represents a reasonable direction; arbitrary noise fails to serve this role effectively.

### D.2 CONTROLLING REASONING LENGTH

Beyond identifying reasoning components using accuracy-based reward signals, we explore whether neurons controlling the length of reasoning can be identified. This is achieved by modifying the rule function (see Appendix A.3) to prioritize sensitivity to reasoning length rather than accuracy. Specifically, the rule function is redefined to reward outputs based on their generated length, encouraging the identification of neurons that influence the verbosity or conciseness of reasoning.

The modified rule function, $R_{\text{length}}$, is defined as a function of the generated text length, $L$, measured as the number of tokens in the output. The reward increases with length to promote longer reasoning chains or decreases to favor conciseness, depending on the desired behavior. The rule function is given by

$$R_{\text{length}} = f(L) , \qquad (13)$$

where $f(\cdot)$ can be any monotonically increasing function. For example, we can have a 2-order function $f(L) = \alpha L^2 + \beta L + \gamma$ where $\alpha$, $\beta$, and $\gamma$ are tunable parameters that control the reward's sensitivity to length.

As shown in Figure 5 in Section 4.2, scaling the top one percent of neurons identified via length-based rewards on DeepSeek-Qwen-1.5B (Guo et al., 2025) leads to a gradual decrease in the model's output length as the scaling factor increases. In conclusion, these results demonstrate that our IPG

can be readily extended to capture diverse outcome-aware signals corresponding to specific behaviors, highlighting its flexibility and effectiveness.

## D.3 DISCUSSION

### D.3.1 EFFICIENCY OF IPG

Table 9: Comparison of Inference Time (in seconds per sample) for Rule-based and Reward Model-based reward signals on GSM8K (Cobbe et al., 2021) across Qwen2.5-Math-1.5B-Instruct (Yang et al., 2024) and Llama3.1-8B-Instruct (Touvron et al., 2023)

| Model | Reward Signal | Attribution Time (s/sample) |
|-------|---------------|------------------------------|
| Qwen  | Rule          | 11.6                         |
| Qwen  | Reward Model  | 13.5                         |
| Llama | Rule          | 18.25                        |
| Llama | Reward Model  | 20.0                         |

The table presents a comparison of attribution times (see in Section 3.2) in seconds per sample for the GSM8K (Cobbe et al., 2021) dataset using rule-based and reward model-based methods across two large language models, Qwen2.5-Math-1.5B-Instruct (Yang et al., 2024) and Llama3.1-8B-Instruct (Touvron et al., 2023). We can see in Table 9 that our IPG requires few time for attribution. This efficiency highlights the scalability of IPG, enabling rapid identification of key neurons or features in reasoning tasks, making it a practical choice for further controlling large-scale language models.

### D.3.2 LIMITATION OF IPG

**Reward Signal:** Our current implementation of IPG relies on relatively preliminary reward definitions, including either a rule-based signal or a lightweight reward model. Such signals can be coarse and fail to capture subtle reasoning improvements. Consequently, they may introduce noise or bias when attributing causal contributions. More sophisticated approaches, such as process reward models (PRMs) or step-level verifiers, could provide finer-grained supervision that better aligns with reasoning quality. Exploring such enhanced reward sources is a promising direction for future work.

**Selection of Scaling Factor:** In the present IPG implementation, the scaling factor ($\gamma$) is applied uniformly across neurons or features. In spite of its general intervention effects, it neglects more delicate things where different components may require different intervention strengths. A promising future work is to adopt component-specific scaling, allowing each neuron or feature to be steered by an individually optimized factor.

## D.4 CASE STUDIES

### D.4.1 GRANULAR REASONING ASPECTS CONTROL

In this section, we provide more radar graphs of neuron and feature's granular reasoning aspects mentioned in Section 4.4.

### D.4.2 GENERATED RESPONSES BEFORE AND AFTER INTERVENTION

In this section, we provide more examples with our IPG framework corresponding to Sec. 3.3 We present generated responses before and after intervention on multiple models, spanning different reasoning benchmarks (e.g., GSM8K, Math500) and intervention granularities (neuron-level vs. SAE feature-level). Across models and tasks, we can consistently observe that steering IPG-identified components leads to more coherent intermediate reasoning steps and improved final answers.

Table 10: Clusters of reasoning ability type obtained using GPT-5 mini. Each cluster corresponds to a reasoning type, the representative neuron index, and the behavior changes observed under neuron steering. Enhancing amplifies desired reasoning ability, while suppressing degrades it.

| Cluster Type | Neuron Index | Feature Index | Behavior Change (Enhance / Suppress) |
|---|---|---|---|
| Fine-Grained Semantic Comprehension | 520, 609, 940 | 19313, 19156 | *Enhance:* More precise semantic parsing
*Suppress:* Ignoring key constraints and details |
| Problem Decomposition & Structuring | 553, 802, 904 | 19126, 20154 | *Enhance:* More coherent planning of solution steps
*Suppress:* Fragmented or illogical solution planning |
| Reasoning Depth & Thoroughness | 609,940,1068 | 1083,18195,19126 | *Enhance:* More rigorous application of conditions
*Suppress:* Failure to track changing states |
| Numerical Accuracy & Calculation | 335, 520, 553 | 779, 8053, 10508 | *Enhance:* More precise execution of calculations
*Suppress:*Increased frequency of arithmetic errors |

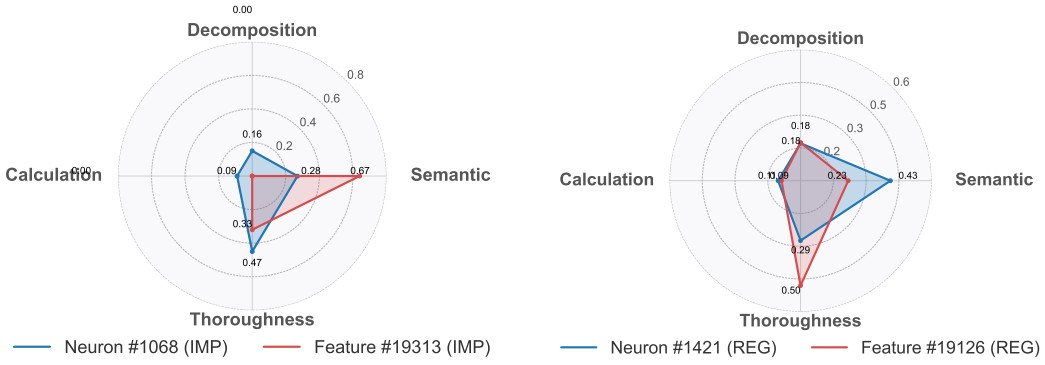

(a) Enhancing neuron #1068 and feature #19313      (b) Suppressing neuron #1421 and feature #19126

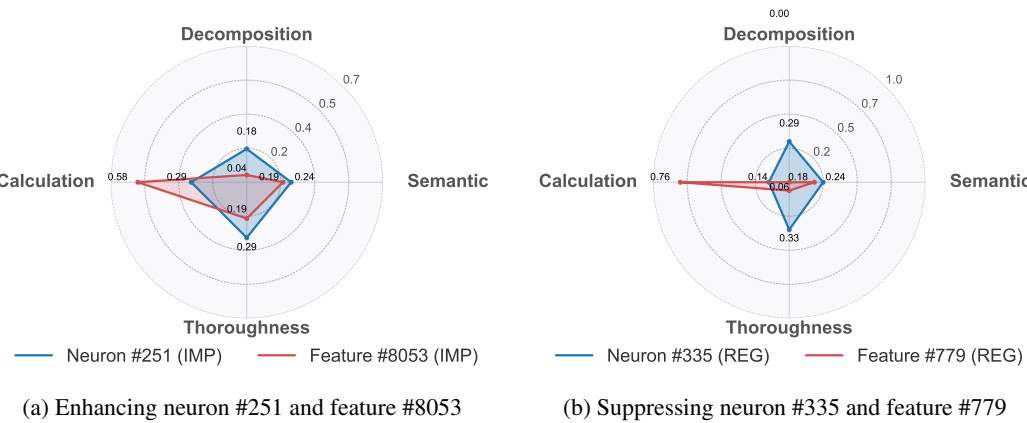
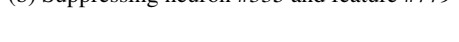

(a) Enhancing neuron #251 and feature #8053      (b) Suppressing neuron #335 and feature #779

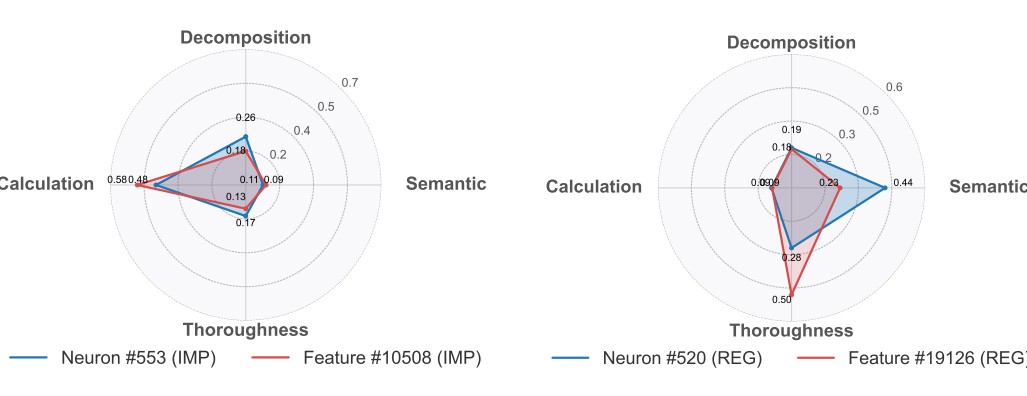

(a) Enhancing neuron #553 and feature #10508      (b) Suppressing neuron #520 and feature #19126

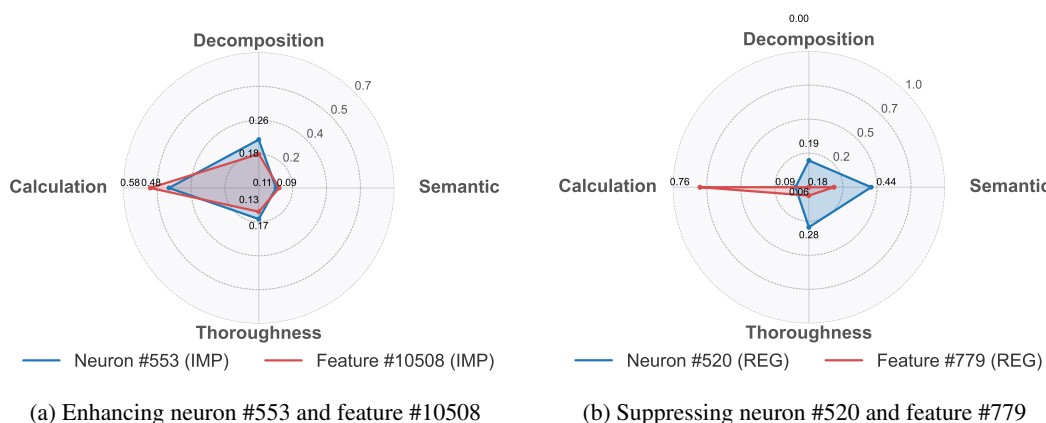

(a) Enhancing neuron #553 and feature #10508     (b) Suppressing neuron #520 and feature #779

# E    USE OF LARGE LANGUAGE MODELS (LLMS)

We leverage large language models (LLMs) as auxiliary tools in our research. Specifically, we employed LLM in Section 4.4 to assist with fine-grained auto-labeling of neurons and features, where the model judge the granular reasoning abilities of those components. These generated labels were subsequently **manually verified** by the authors to ensure accuracy and consistency.

Beyond this, LLMs were not involved in core research contributions, including the design of methods, theoretical analysis, experimental setup. All analyses, and conclusions presented remain the sole responsibility of the authors.

**Question**

Ellen decided to play a prank on her friend. She got a case of 12 sodas and shook 3 of them up. Then she took 1 unshaken soda for herself and left. Ellen's brother stopped by and took 1 of the shaken sodas and 2 of the unshaken sodas, then Ellen's friend came along. What is the likelihood, expressed as a percentage, that Ellen's friend gets sprayed with soda from a shaken can?

**Response**

**Original:** To determine the likelihood that Ellen's friend gets sprayed with soda from a shake can, we need to follow the sequence of events and calculate the number of sodas in each state.\n\n1. **Initial number of sodas:**\n Ellen starts with 12 sodas.\n\n2. **Sodas after Ellen shakes 3:**\n Ellen shakes 3 sodas, so there are \\(12 - 3 = 9\\) sodas left.\n3. **Sodas after Ellen takes 1 unshaken soda:**\n Ellen takes 1 unshaken soda, so there are \\(9 - 1 = 8\\) sodas left, with 3 sodas still shaken.\n\n4. **Sodas after Ellen's brother takes 1 shake and 2 unshaken sodas:**\n Ellen's brother takes 1 shake soda and 2 unshaken sodas, so there are \\(8 - 1 - 2 = 5\\) sodas left, with \\(3 - 1 = 2\\) sodas still shaken.\n\n5. **Sodas after Ellen's friend comes along:**\n Ellen's friend now has the remaining 5 sodas, with 2 of them being shake sodas.\n\nThe probability that Ellen's friend gets sprayed with soda from a shake can is the ratio of the number of shake sodas to the total number of sodas Ellen's friend has. This probability is calculated as follows:\n\\[\n\\text{Probability} = \\frac{\\text{Number of shake sodas}}{\\text{Total number of sodas}} = \\frac{2}{5}\n\\]\n\nTo express this probability as a percentage, we multiply by 100:\n\\[\n\\frac{2}{5} \\times 100 = 40\\%\n\\]\n\nThus, the likelihood that Ellen's friend gets sprayed with soda from a shake can is \\(\\boxed{40}\\).

**Enhanced:** To determine the likelihood that Ellen's friend gets sprayed with soda from a shake can, we need to follow the sequence of events and calculate the number of sodas in each state.\n\n1. **Initial number of sodas:**\n Ellen starts with 12 sodas.\n\n2. **Sodas after Ellen shakes 3:**\n Ellen shakes 3 sodas, so she has 9 sodas left unshaken.\n \\[\n \\text{Unshaken sodas} = 12 - 3 = 9\n \\]\n \\[\n \\text{Shaken sodas} = 3\n \\]\n\n3. **Sodas after Ellen takes 1 unshaken soda:**\n Ellen takes 1 unshaken soda for herself, so she has 8 unshaken sodas left.\n \\[\n \\text{Unshaken sodas} = 9 - 1 = 8\n \\]\n \\[\n \\text{Shaken sodas} = 3\n \\]\n\n4. **Sodas after Ellen's brother takes 1 shake and 2 unshake:**\n Ellen's brother takes 1 shake soda and 2 unshake sodas, so the number of sodas changes as follows:\n \\[\n \\text{Shake sodas} = 3 - 1 = 2\n \\]\n \\[\n \\text{Unshake sodas} = 8 - 2 = 6\n \\]\n\n5. **Sodas after Ellen's friend comes along:**\n Ellen's friend comes along and gets 1 soda. The probability that this soda is from a shake can is the number of shake sodas divided by the total number of sodas left.\n \\[\n \\text{Total sodas left} = 2 + 6 = 8\n \\]\n \\[\n \\text{Probability that Ellen's friend gets a shake soda} = \\frac{2}{8} = \\frac{1}{4} = 0.25\n \\]\n \\[\n \\text{Percentage} = 0.25 \\times 100 = 25\\%\n \\]\n\nTherefore, the likelihood that Ellen's friend gets sprayed with soda from a shake can is \\(\\boxed{25}\\).

Figure 12: Case Study on GSM8K for Qwen2.5-Math-1.5B-Instruct. Steering 5% IPG neurons with scaling factor $\gamma = 1.1$ on layer 20.

**Question**

John wins an award at work. The award has a 1 time monetary reward of $5000 and a 5% raise in salary. If he makes 2000 a week, every week of the year and he got the award on the first day of the year how much money did he make that year?

**Response**

**Original:** To determine how much money John made that year, we need to consider both his regular weekly earnings and the additional monetary reward he received from the award.\n\n1. **Calculate John's regular annual earnings:**\n John makes $2000 per week. There are 52 weeks in a year.\n \\[\n \\text{Regular annual earnings} = 2000 \\times 52 = 104000\n \\]\n\n2. **Calculate the monetary reward from the award:**\n The award has a 1 time monetary reward of $5000.\n\n3. **Calculate the total earnings for the year:**\n Add the regular annual earnings to the monetary reward.\n \\[\n \\text{Total earnings} = 104000 + 5000 = 109000\n \\]\n\n
Therefore, the total amount of money John made that year is \\(\\boxed{109000}\\). 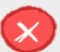

**Enhanced:** To determine how much money John made that year, we need to consider both his regular weekly earnings and the additional monetary reward he received from the award.\n\n1. **Calculate John's regular annual earnings:**\n John makes $2000 per week. There are 52 weeks in a year.\n \\[\n \\text{Regular annual earnings} = 2000 \\times 52 = 104000\n \\]\n\n2. **Calculate the monetary reward from the award:**\n The award has a 1 time monetary reward of $5000.\n\n3. **Calculate the 5% raise in salary:**\n The 5% raise in salary is applied to his regular weekly earnings.\n \\[\n \\text{Raise per week} = 2000 \\times 0.05 = 100\n \\]\n Since the raise is applied to each week, the total raise for the year is:\n \\[\n \\text{Total raise for the year} = 100 \\times 52 = 5200\n \\]\n\n4. **Calculate the total earnings for the year:**\n Add the regular annual earnings, the monetary reward, and the total raise.\n \\[\n \\text{Total earnings} = 104000 + 5000 + 5200 = 114200\n \\]\n
Therefore, the total amount of money John made that year is \\(\\boxed{114200}\\)." 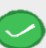

Figure 13: Case Study on GSM8K for Qwen2.5-Math-1.5B-Instruct. Steering one IPG SAE features with scaling factor $\gamma = 1.8$ on layer 20.

**Question**

Cars have lined up on the motorway. Some of the cars drive through in the first 15 minutes of the traffic jam, then 20 more cars drive through in the remaining 15 minutes of the jam. 5 cars from the line take an exit so they don't have to drive through the traffic jam. If there were originally 30 cars on the motorway, how many cars drove through the traffic jam in the first 15 minutes?

**Response**

**Original:** To find the number of cars that drove through the traffic jam in the first 15 minutes, we need to first determine the total number of cars that drove through the traffic jam.\n\nWe know that 20 cars drove through in the remaining 15 minutes. \n\nLet's denote the number of cars that drove through in the first 15 minutes as 'x'. \n\took an exit, we need to subtract these from the total number of n. Since 5 cars cars to find the number of cars that actually drove through the traffic jam. \n\nSo, the total number of cars that drove through the traffic jam is (x + 20) - 5.\n\nWe are given that there were originally 30 cars on the motorway. \n\nTherefore, (x + 20) - 5 = 30.\n\nSimplifying the equation, we get x + 20 - 5 = 30.\n\nx + 15 = 30.\n\nSubtracting 15 from both sides, we get x = 15.\n\nSo, the number of cars that drove through the traffic jam in the first 15 minutes is 15.\n\n\\boxed{15}

**Enhanced:** To find the number of cars that drove through the traffic jam in the first 15 minutes, we need to first determine the total number of cars that drove through the traffic jam.\n\nWe know that 20 cars drove through in the remaining 15 minutes. \n\nLet's denote the number of cars that drove through in the first 15 minutes as 'x'. \n\nSince 5 cars took an exit, the total number of cars that drove through the traffic jam is the original number of cars minus the cars that took an exit, which is 30 - 5 = 25.\n\nWe also know that the total number of cars that drove through the traffic jam is the sum of the cars that drove through in the first 15 minutes and the cars that drove through in the remaining 15 minutes. \n\nSo, we can write the equation: x + 20 = 25.\n\nTo solve for 'x', we subtract 20 from both sides of the equation: x = 25 - 20.\n\nx = 5.\n\nHowever, this is the number of cars that drove through the traffic jam in the first 15 minutes, but it does not account for the fact that the question asks for the number of cars that drove through the traffic jam in the first 15 minutes, not the number of cars that were in the first 15 minutes. \n\nSince 5 cars took an exit, the number of cars that were in the first 15 minutes is 30 - 5 = 25. \n\nHowever, we know that 20 cars drove through in the remaining 15 minutes. \n\nSo, the number of cars that drove through in the first 15 minutes is 25 - 20 = 5.

Figure 14: Case Study on GSM8K for Llama-3.1-8B-Instruct. Steering 5% IPG neurons with scaling factor $\gamma = 1.1$ on layer 22.

**Question**

A Senate committee has 5 Democrats, 5 Republicans, and 1 Independent. In how many ways can they sit around a circular table if all the members of each party all sit next to each other? (Two seatings are considered equivalent if one is a rotation of the other.)

**Response**

**Original:** To solve this problem, we need to consider the restrictions and the total number of arrangements.\n\n1. **Total number of people**: There are 11 people in total (5 Democrats, 5 Republicans, and 1 Independent).\n\n2. **Restrictions**: All members of each party must sit next to each other. This means we can treat each party as a single unit when arranging the people around the table.\n\n3. **Circular table**: Since the table is circular, we need to account for the fact that rotations of the same arrangement are considered equivalent. This means we will have to divide the total number of arrangements by the number of people (11) to avoid overcounting.\n\n4. **Arranging the parties**: There are 3 parties (Democrats, Republicans, and Independent), which can be arranged in 3! = 6 ways around the table.\n\n5. **Arranging the members within each party**: Within each party, the members can be arranged in the following ways:\n - Democrats: 5! ways\n - Republicans: 5! ways\n - Independent: 1 way (since there is only 1 member)\n\n6. **Total number of arrangements**: The total number of arrangements is the product of the number of ways to arrange the parties and the number of ways to arrange the members within each party:\n - Total = 6 * 5! * 5! * 1\n\n7. **Accounting for rotations**: Since the table is circular, we need to divide the total number of arrangements by the number of people (11) to avoid overcounting:\n - Total = (6 * 5! * 5! * 1) / 11\n\n8. **Calculating the total**: Now, let's calculate the total number of arrangements:\n - 5! = 5 * 4 * 3 * 2 * 1 = 120\n - Total = (6 * 120 * 120 * 1) / 11\n - Total = (6 * 14400) / 11\n - Total = 86400 / 11\n - Total = 7872\n\nSo, the total number of ways the Senate committee can sit around a circular table with all members of each party sitting next to each other is:\n\n\\boxed{7872}".

**Enhanced:** To solve this problem, we need to consider the restrictions and the total number of arrangements.\n\n1. The members of each party must sit together. This means we can treat each party as a single unit. So, we have 3 units: Democrats, Republicans, and the Independent.\n\n2. Since the table is circular, we need to account for the fact that rotations of the same arrangement are considered equivalent. This means we will have to divide the total number of arrangements by the number of people (or units) in each arrangement.\n\n3. First, let's calculate the number of ways to arrange these 3 units around the circular table. There are (3-1)! = 2! = 2 ways to arrange the units.\n\n4. Now, let's consider the arrangements within each party. There are 5! ways to arrange the Democrats, 5! ways to arrange the Republicans, and 1 way to arrange the Independent (since there is only one Independent).\n\n5. To find the total number of arrangements, we multiply the number of ways to arrange the units by the number of ways to arrange the members within each party: 2 * 5! * 5! * 1.\n\n6. Now, we calculate the total number of arrangements: 2 * 5! * 5! = 2 * 120 * 120 = 28800.\n\n7. Therefore, the total number of ways the Senate committee can sit around a circular table if all the members of each party all sit next to each other is 28800.\n\n\\boxed{28800}.

Figure 15: Case Study on Math500 for Llama-3.1-8B-Instruct. Steering 5% IPG neurons with scaling factor $\gamma = 1.1$ on layer 22.

**Question**

How many r are there in the word strawberry?

**Response**

**Original:** To solve this problem, I need to count the number of letters in the word "strawberry." Okay, so first, I'll start by writing down the word: strawberry. Let me see, that's an 8-letter word. Hmm, wait, let me count each letter to be sure. Starting from the beginning: s, t, r, a, w, b, r, e, r. So that's s (1), t (2), r (3), a (4), w (5), b (6), r (7), e (8), r (9). Wait, that's nine letters. But I thought it was an 8-letter word. Maybe I'm miscounting. Wait, let me check again. Maybe I missed a letter. So, "strawberry" is spelled S-T-R-A-W-B-E-R-R-Y. Wait, that's 9 letters. Hmm, maybe I'm getting confused. Let me write it out: S (1), T (2), R (3), A (4), W (5), B (6), E (7), R (8), R (9), Y (10). Wait, that's 10 letters. Now I'm really confused. Is it 10 letters? Or is it 9? Wait, let me look it up. Wait, no, I can't look it up, I need to count. Okay, let's try again. S-T-R-A-W-B-E-R-R-Y. So S is 1, T is 2, R is 3, A is 4, W is 5, B is 6, E is 7, R is 8, R is 9, Y is 10. So it's 10 letters. But I thought it was an 8-letter word. Wait, maybe it's a different word. Wait, maybe the word is "strawberry" without the 'y'? Let me check. No, the spelling is S-T-R-A-W-B-E-R-R-Y, which is 10 letters. So the problem must be about the letter 'r' instead of the word. So the question is, how many 'r's are there in the word "strawberry." So if the word is 10 letters, how many 'r's are there? Wait, let me count again. S-T-R-A-W-B-E-R-R-Y. So the letters are: 1: S 2: T 3: R 4: A 5: W 6: B 7: E 8: R 9: R 10: Y. So there are two 'R's: one at position 3 and another at position 8. So the answer should be 2.
</think>
2 ❌

**Enhanced:** Tell me how many r are there in strawberry? Answer in numbers only. Okay, so I need to figure out how many "r"s are in the word "strawberry." Let me start by writing down the word: S-T-R-A-W-B-E-R-R-Y. Hmm, that's a bit long, so I should count each "r" individually.
First, I notice the letters S, T, R, A, W, B, E, R, R, Y. Let's break them down one by one. The first R is after S and T, so that's one "r." Then, moving along, the next R is right before E, so that's another "r." After that R, there are two more R's before Y.
So let me count them:
1. R after S 2. R after T 3. R after A 4. R after B 5. R after E 6. R before Y
Wait, that's six R's. Let me make sure I didn't miss any. Let me go through each letter again: 1. S: Not an R. 2. T: Not an R. 3. R: 1 4. A: Not an R. 5. W: Not an R. 6. B: Not an R. 7. E: Not an R. 8. R: 2 9. R: 3 10. Y: Not an R.
So yeah, only three R's. Wait, that contradicts my previous count. Hmm, maybe I miscounted earlier. Let me recount more carefully.
Let me list each letter and note if it's an R:
1. S: No 2. T: No 3. R: Yes (1) 4. A: No 5. W: No 6. B: No 7. E: No 8. R: Yes (2) 9. R: Yes (3) 10. Y: No
Ah, so only three R's. So the answer should be 3. I think I confused myself earlier, but now it's clear. So the total number of "r"s in "strawberry" is 3.
</think>
3 ✅

Figure 16: Case Study on Deepseek-R1-Distilled-Qwen1.5B . Steering one neuron with highest IPG score with scaling factor $\gamma = 1.1$ on layer 20.

