# OpenReview forum: "Interpreting and Controlling LLM Reasoning through Integrated Policy Gradient"
_ICLR.cc/2026/Conference — Submitted to ICLR 2026_

### Official Review · Reviewer_2ZEq · 2025-10-27

**Soundness:** 3
**Presentation:** 3
**Contribution:** 3
**Rating:** 4
**Confidence:** 3

**Summary:**

This paper introduces Integrated Policy Gradient (IPG), a method intended to attribute and modulate reasoning components in large language models by applying a policy‐gradient–like formulation on hidden activations, followed by scaling of the identified components. The aim is to locate “reasoning circuits” and then control them via a scalar $\gamma$. Experiments on math‐reasoning datasets (GSM8K, MATH500, AIME2024, GPQA‐Diamond) show modest gains.

While the aim is timely, the paper falls short of clearly distinguishing itself from recent work in RL‐based reasoning and lacks methodological and empirical depth to justify its claims of mechanistic interpretability.

**Strengths:**

1. Addresses an important goal — locating reasoning features inside LLMs and offering intervention methods rather than purely black‐box fine-tuning.
2. Builds on long‐horizon rewards rather than purely prediction loss, which is underexplored in hidden‐state attribution.
3. The empirical gains shown (some improvement when scaling selected neurons) at least suggest there is signal in the method.

**Weaknesses:**

1. IPG’s experiments manipulate hidden states (Eq.~(4) scaling), which is more akin to an optimization/steering method rather than a genuine causal attribution. Results resemble those from EM‐PG (arXiv:2504.18587), which focus on optimizing reasoning behaviours rather than tracing hidden “circuits”.

2.  Eq.~(3)’s selection statistic \(S_i\) aggregates across samples without variance control or normalization. Considering the off‐policy nature of PG in RPG, how is sample bias handled? The more rigorous treatment in RPG (stop‐gradient, importance weights) suggests IPG is under‐specified.

3. IPG’s transfer improvements are minor (< +1.2%), whereas RPG achieved up to +6 pp in comparable settings. Without clear generalisation, the claimed “general reasoning circuit” is weak.

4. No mediation analysis, no circuit dissection across heads/layers, no intervention experiments that isolate a sub‐module and show step‐wise effect. By contrast, interpretability‐driven works emphasise causality and modularity. IPG’s approach remains correlation‐based.

5. Given EM‐PG and RPG already apply policy‐gradient frameworks to LLM reasoning, the novelty of applying PG to hidden states (rather than parameters) needs stronger justification.

**Questions:**

1. In Eq.~(1), you write \(\frac{\partial J(h)}{\partial h} = \mathbb{E}_{\tau\sim\pi_\theta} \left[ \frac{\partial \log \pi_\theta(\tau; h)}{\partial h} (R(\tau)-b) \right]\). But since \(\pi_\theta\) is parameterised by \(\theta\), and \(h\) is a deterministic hidden activation rather than stochastic policy variable, can you provide a rigorous derivation from the log‐derivative trick?

2. Regarding Eq.~(2)’s path integral, how do you select the baseline \(h'\)? Have you tested sensitivity to different baselines (zero vector, mean hidden state, random noise)?

3.For the selection score \(S_i\) in Eq.~(3), why is simple averaging used? Why not adopt variance‐based or Shapley‐value–style attribution for hidden features?

4. In Eq.~(4), if you set \(\gamma\to\infty\), does performance keep improving? If so, then you are simply amplifying hidden signals globally, which undermines the claim of localised reasoning features.

5. How does your method handle off‐policy bias and importance sampling? In RPG (arXiv:2505.17508) the weighting of importance samples is a central issue; what safeguards does IPG include?

6. You compare to baseline LLMs and “vanilla” fine‐tuning, but did you evaluate against RPG or EM‐PG methods directly? If not, why?

7. Please provide trial‐to‐trial variance (across seeds) and statistical significance of intervention effects, especially in the suppression experiments where accuracy drops to zero—how generalisable are these results?

8. Can you show a case where your intervention corrects an incorrect reasoning chain (intermediate step), not just the final answer? This is crucial for interpretability claims.

9. Since I am very interested in your work, but I have seen some other interesting methods on Google Scholar (not referring to similarity), I think if you could give me a concise comparative explanation, I think it would be very convincing. (arXiv:2505.17508)(arXiv:2504.18587)

---

> ### Author Response · Authors · 2025-11-22
> **Rebuttal by Authors**
>
> Dear reviewer 2ZEq, thank you for your comments and thoughtful questions. Following is our response.
> > W1. IPG’s experiments manipulate hidden states (Eq.~(4) scaling), which is more akin to an optimization/steering method rather than a genuine causal attribution. Results resemble those from EM‐PG (arXiv:2504.18587), which focus on optimizing reasoning behaviours rather than tracing hidden "circuits".
>
> Thank you for your comments.
> We would like to clarify that our method is different from the optimization work like EM-PG [2] you mentioned, the differences are three-fold as detailed as below.
> * **Task difference**: IPG is a gradient-based, training-free *interpretability* method targeting at reasoning mechanism of Large Language Models (LLMs). It leverages policy gradient to measure the contribution of internal components such as the neuron within the hidden representation and identifies the most influential set of components. Optimization methods such as EM-PG [2] you mentioned are to finetune the LLM, affecting the whole model parameters, not to interpret the model.
> * **Methodology difference**: IPG applies gradient-based attribution method and aggregates policy gradient along the path from a baseline (e.g. zero value) to the observed activation level of each component (e.g. neuron) to attribute reasoning behavior. Optimization methods such as EM-PG [2] you mentioned frame reasoning in an LLM as an Expectation-Maximization (EM) optimization problem, including sampling diverse trajectories from the previous iteration's model and fine-tuning the LLM via reward-guided updates. The design and goal of the methodologies differ significantly from each other.
> * **Experiment difference**: IPG experiment settings are designed from an *interpretability* perspective. IPG uses scaling method to steer on the identified components (neurons or features) in the model's reasoning behavior to validate that we have accurately identified the components, and we compare with interpretability methods.  Besides, we also conduct experiments like granular reasoning ability analysis (subability such as calculation) by manipulating one single component. By contrast, optimization methods such as EM-PG [2] which finetune the model, put an emphasis on improving overall performance (e.g. accuracy).
>
> > W2. Eq.(3)’s selection statistic $(S_i)$ aggregates across samples without variance control or normalization. Considering the off‐policy nature of PG in RPG, how is sample bias handled? The more rigorous treatment in RPG (stop‐gradient, importance weights) suggests IPG is under-specified.
>
> Thank you for raising this important point. We make clarification below.
> Off-policy concerns do not apply to our IPG attribution. **All computations are *strictly on-policy***, and the gradient estimate has no off-policy sample bias. In reinforcement learning, off-policy bias occurs when samples generated by one policy are used to estimate gradients for another, updated policy. By contrast, IPG operates entirely on a *single, fixed* model (the model being interpreted) with no parameters updates. Every computation in IPG, including generation and policy-gradient estimation, is performed on this model.
>
> > W3. IPG’s transfer improvements are minor (< +1.2%), whereas RPG achieved up to +6 pp in comparable settings. Without clear generalisation, the claimed "general reasoning circuit" is weak.
>
> Thanks for pointing this out. We would like to clarify that the transfer improvements are reasonable.
> (i) **Goal of the transfer experiment:** The transfer setting in Section 4.3 evaluates whether IPG accurately identifies general reasoning components that remain effective when moved to another benchmark. Under this goal, IPG consistently outperforms all **interpretability baselines** (Table 2 in our paper), demonstrating stronger cross-benchmark generalization.
> (ii) **Difference between interpretability and optimization-based methods:** RPG [1] 's larger gains stem from being an online optimization method that updates parameters and adapts during training. In contrast, IPG is an interpretability-based method as it identifies and manipulates fixed components without training or extra supervision. Such methods typically yield smaller absolute performance gains, but they offer mechanistic insight and explicit controllability, which optimization methods do not. By the way, we claim "general reasoning mechanism" instead of "general reasoning circuit".

---

> ### Author Response · Authors · 2025-11-22
> **Rebuttal by Authors**
>
> > W4. No mediation analysis, no circuit dissection across heads/layers, no intervention experiments that isolate a sub-module and show step-wise effect. By contrast, interpretability-driven works emphasise causality and modularity. IPG’s approach remains correlation-based.
>
> We clarify the scope and position of our method in the following three aspects:
> (i) **Scope of our claim is on "components" instead of "circuits".** We do not claim to recover full "circuits". IPG identifies internal components (neurons or SAE features) in LLMs that play causal roles in reasoning behavior. Our method isolates influential subsets of components rather than reconstructing head-to-layer pathway structures.
> (ii) **IPG is a causal attribution method, not correlation-based:** IPG directly measures the causal effect of components by aggregating policy gradients along the path from a baseline to the observed activation level of each component, which follows the standard formulation of causal mediation analysis (CMA). As summarized in a CMA survey [6], the "mediator" can be neurons, feature sets, or larger circuits and our work instantiates this paradigm at the neuron or feature level.
> (iii) **Relation to circuit-level analysis:** Circuit-level mediation or cross-layer dissection is complementary to our approach. IPG provides a component-level causal interpretation framework for reasoning behavior, which can serve as a building block for more structured circuit analysis. Extending IPG to identify multi-layer causal pathways is a promising future direction.
>
> > W5. Given EM-PG [2] and RPG [1] already apply policy-gradient frameworks to LLM reasoning, the novelty of applying PG to hidden states (rather than parameters) needs stronger justification.
>
> We will clarify the novelty of IPG relative to EM-PG [2] and RPG [1] from three aspects:
> (i) **IPG leverages policy-gradient for attribution, not as an optimization method.** EM-PG [2] and RPG [1] develop improved PG algorithms for training models. In contrast, IPG does not update parameters. We repurpose the PG objective as a causal attribution measure over hidden-state components. This lets us quantify the cumulative and long-range contribution of neurons or features to reasoning behavior that existing **interpretability methods** cannot capture.
> (ii) **IPG integrates PG with gradient-based attribution to overcome limitations of prior interpretability work.** Standard gradient-based attribution methods struggle with sparse, non-differentiable outcomes (e.g., final-answer correctness in reasoning tasks). IPG's provides attribution of cumulative, long-range effects, overcoming challenges of sparse and non-differentiable outcome signals by further incorperating PG with integrated path accumulation.
> (iii) **There exists conceptual difference in goal**. EM-PG [2] and RPG [1] aim to improve reasoning performance via online optimization. IPG aims to understand the mechanism that drives reasoning behavior and to identify causally influential components. These are fundamentally different problem settings.

---

> ### Author Response · Authors · 2025-11-22
> **Rebuttal by Authors**
>
> > Q1. In Eq.~(1), $\frac{\partial J(h)}{\partial h} = \mathbb{E}\_{\tau\sim\pi_\theta} \left[ \frac{\partial \log \pi_\theta(\tau; h)}{\partial h} (R(\tau)-b) \right]$, $\pi_\theta$ $\pi$ is parameterised by $\theta$, and $h$ is a deterministic hidden activation rather than stochastic policy variable, can you provide a rigorous derivation from the log‐derivative trick?
>
> Thank you for pointing this out. We show the full derivation for the continuous case below.
> $$J(h) = \int \pi_\theta(\tau; h) R(\tau) \, d\tau.$$
> Differentiate under the integral.
> $$\frac{\partial J(h)}{\partial h} = \int \frac{\partial \pi_\theta(\tau; h)}{\partial h} R(\tau) \, d\tau.$$
> Apply the log-derivative identity (score function trick).
> $$\frac{\partial \pi_\theta(\tau; h)}{\partial h} = \pi_\theta(\tau; h) \frac{\partial \log \pi_\theta(\tau; h)}{\partial h}.$$
> Substitute into the integral:
> $$\frac{\partial J(h)}{\partial h} = \int \pi_\theta(\tau; h) \frac{\partial \log \pi_\theta(\tau; h)}{\partial h} R(\tau) \, d\tau = \mathbb{E}\_{\tau\sim\pi_\theta} \left[ \frac{\partial \log \pi_\theta(\tau; h)}{\partial h} R(\tau) \right].$$
> For baseline b (independent of $\tau$),
> $$\mathbb{E}\_{\tau\sim\pi_\theta} \left[ \frac{\partial \log \pi_\theta(\tau; h)}{\partial h}\cdot b \right] = b \cdot \int \frac{\partial \log \pi_\theta(\tau; h)}{\partial h} \pi_\theta(\tau; h) \, d\tau .$$
> But $\int \pi_\theta(\tau; h) \, d\tau = 1$ for every $h$, so differentiate that identity w.r.t. $h$:
> $$\frac{\partial}{\partial h} \int \pi_\theta(\tau; h) \, d\tau = \int \frac{\partial \pi_\theta(\tau; h)}{\partial h} \, d\tau = 0.$$
> Hence $\mathbb{E} [\partial \log \pi_\theta / \partial h] \cdot b = 0$. Subtracting this zero from the expression above gives
> $$\frac{\partial J(h)}{\partial h} = \mathbb{E}\_{\tau\sim\pi_\theta} \left[ \frac{\partial \log \pi_\theta(\tau; h)}{\partial h} (R(\tau) - b) \right].$$
> This is exactly the stated equation. Therefore the same format of equation can be applied to hidden states $h$ as in Eq. (1).
>
> > Q2. Regarding Eq.~(2)’s path integral, how to select the baseline (h')? (sensitivity for zero vector, mean hidden state and random noise)
>
> We apologize for any confusion on baseline $h'$.
> (i) We use zero vector in paper.
> (ii) We also add an experiment with "mean hidden state" and "random noise" based on Qwen-Math2.5-1.5B-Instruct on GSM8K dataset. Our IPG is **not sensitive** on choices of different baseline $h'$.
> **Experiment setting**: We report the accuracy metric (i.e. greedy decoding) as mentioned in Section 4.1. "Enhance" measures the accuracy after strengthening the identified reasoning components (higher is better), while "Suppress" measures the rate after weakening them (lower is better).
> **Analysis**: As shown in Table 1 below, we achieve excellent performance under these baselines $h'$, and they do not affect of model control much. "Random noise" shows the worst performance because it does not serve as a reasonable choice for baseline $h'$. We include it only as a non-informative reference.
>
> Table 1: Sensitivity analysis on choices of baseline $h'$. The original (before intervention) accuracy is 82.41\%.
> | Baseline ($h'$) | GSM8K (Enhance $\uparrow$) | GSM8K (Suppress $\downarrow$) |
> |-------|-------|-------|
> | Zero vector | 84.38 |0.0 |
> |  Mean | 84.45 |21.15 |
> | Random noise| 84.00 |78.16|
>
> We have added the sensitivity analysis of baseline $h'$ in Appendix D.1. The revised part has been highlighted  with blue.

---

> ### Author Response · Authors · 2025-11-22
> **Rebuttal by Authors**
>
> > Q4. In Eq.~(4), if you set $\gamma\to\infty$, does performance keep improving? If so, then you are simply amplifying hidden signals globally, which undermines the claim of localised reasoning features.
>
> Thank you for the question.
> (i) **Increasing $\gamma$ does not yield monotonic improvement.** As shown in Fig. 2(a), performance initially increases but then drops when $\gamma$ becomes too large. This is also consistent with prior interpretability works [3,4], which similarly report a peak followed by degradation.
> (ii)  **$\gamma$ is not used to globally amplify hidden signals.** The reason is that a neuron's activation has a valid operational range. Scaling by a moderate $\gamma$ amplifies the intended effects, but scaling too aggressively pushes hidden states outside the model's manifold, destabilizing downstream computations. Thus, $\gamma$ is a controlled intervention that only strengthens the contribution of the selected localized components within a reasonable range.
>
> > Q5. How does your method handle off‐policy bias and importance sampling? In RPG [1] the weighting of importance samples is a central issue; what safeguards does IPG include?
>
> IPG does not suffer from off-policy bias, as stated in W2.
> (i) **Off-policy bias does not arise in IPG:** It appears in reinforcement learning settings when gradients are computed from samples generated by one policy but used to update a different policy. In contrast, IPG operates entirely on a *single, fixed* model (the model being interpreted) with no parameters updates.
> Every computation in IPG, including generation and policy-gradient estimation, is performed on this model. Therefore, all computations are *strictly on-policy*, and the gradient estimate has no off-policy sample bias.
> (ii) In RPG [1], safeguards such as importance sampling are required because RPG updates the policy using samples generated by a different policy, which introduces off-policy bias. In contrast, IPG does not require these safeguards, because IPG does not perform policy updates and does not mix behavior and target distributions. All computations in IPG, including generation, attribution, and score aggregation, are carried out on a single fixed model.
>
> > Q6. The paper compares to baseline LLMs and "vanilla" fin-tuning, but why do not evaluate against RPG [1] or EM‐PG [2] methods directly?
>
> Thank you for the question. We clarify that we do not compare "vanilla" fine-tuning and why RPG [1] and EM-PG [2] are not included as baselines below.
> (i) **IPG is a training-free interpretability method**, not a policy-gradient optimization algorithm. Although IPG incorporates a policy-gradient formulation, it uses it only for causal attribution, not for training. RPG [1] and EM-PG [2], in contrast, are policy-gradient training algorithms that update model parameters to improve reasoning accuracy. Thus, they have a fundamentally different objective and are not directly comparable in our evaluation setting.
> (ii) **Our comparison focuses on interpretability instead of training**. Our goal is to evaluate whether different interpretability methods can identify the correct internal components (neurons or features) by intervening on them and observing behavioral changes. Therefore, we compare against interpretability-based baselines rather than including optimization-based training algorithms.
>
> > Q7. Please provide trial-to-trial variance (across seeds) and statistical significance of intervention effects, especially in the suppression experiments where accuracy drops to zero-how generalisable are these results?
>
> A7. Thank you for your feedback on the experimental evaluation. We calculate p-value w.r.t. Activation baseline and demonstrate it (as it shows consistent performance among baselines) in Table 5 below.
> We can observe that almost all p-values are **below the established 0.05 significance level**, which provides robust statistical evidence affirming our IPG's superiority when compared to baselines. The p-value for AIME-2024 is higher as the benchmark only contains few samples (30) and we achieve similar performance with Activation. The benchmark's limited size restricts statistical power.
>
> Table 5: P-Values (in parentheses) of accuracy vs Activation baseline with Qwen2.5-Math-1.5B-Instruct.
> | Benchmark | Enhance: (Ours)| Enhance: (Activation) | Suppress: (Ours) | Suppress: (Activation) |
> |-|-|-|-|-|
> | **GSM8K**        |84.38 (0.024) |     82.86 (0.024)   | 0.00 (0.000) | 67.78 (0.000)
> | **MATH-500**     | 64.60 (0.023)|    63.60 (0.023)  | 1.60 (0.000) | 39.80 (0.000)
> | **AIME-2024**    |   20.00 (0.617)    | 20.00 (0.617)   | 0.00 (1.000)| 0.00 (1.000)
> | **GPQA-Diamond** |   30.30 (0.009)    |       25.75 (0.009)      | 0.00 (1.37e-6) | 15.15 (1.37e-6)

---

> ### Author Response · Authors · 2025-11-22
> **Rebuttal by Authors**
>
> > Q8. Can you show a case where your intervention corrects an incorrect reasoning chain (intermediate step), not just the final answer? This is crucial for interpretability claims.
>
> We have provided some cases studies in Appendix D. For example, in Figure 14, we have such an case study on GSM8K for Llama-3.1-8B-Instruct. We present the snippet  below:
> **Before steering** (incorrect reasoning chain). The model skips a key intermediate step (computing the 5% raise) and therefore produces the wrong final answer:
> ```text!
> To determine how much money John made that year ... 1.Calculate John's regular annual earnings: ... 104000\n 2.Calculate the monetary reward from the award:\n ... $5000.\n (**miss key intermediate reasoning step**) 3.Calculate the total earnings for the year: ... 109000\n \n\nTherefore, the total amount of money John made that year is 109000. (**wrong answer**)
> ```
> **After IPG steering** (corrected reasoning chain).
> Our intervention restores the missing reasoning step, enabling the model to complete a coherent multi-step chain:
> ```text!
> To determine how much money John made that year, ... 1. Calculate John's regular annual earnings: ... 104000\n 2. Calculate the monetary reward from the award:\n ... $5000.\n 3. Calculate the 5% raise in salary: (**complete intermediate reasoning step**) ... 5200\n 4. **Calculate the total earnings for the year: ... 114200\n Therefore, the total amount of money John made that year is 114200. (**correct answer**)
> ```
> This demonstrates that IPG does not merely push the model toward a correct final output. It also improves the internal reasoning trajectory, making the reasoning more complete and coherent. Additional examples are provided in Appendix A.4.2.
>
> > Q9. Since I am very interested in your work, but I have seen some other interesting methods on Google Scholar (not referring to similarity), I think if you could give me a concise comparative explanation, I think it would be very convincing. (arXiv:2505.17508)(arXiv:2504.18587)
>
> Thanks for your comment. The key distinction is that IPG is an interpretability method, while RPG [1] and EM-PG [2] are optimization methods.
> (i) **Goal difference**:  Our work focuses on interpretability for reasoning. We identify the internal components (neurons or features) that causally drive reasoning behavior. We do not train or update the model. The two papers [1,2] aim at optimization, not mechanistic interpretation, and thus they address a fundamentally different problem.
> (ii) **Methodology difference**: IPG incorporates a policy-gradient-style only as an attribution framework (to measure long-range contribution for reasoning behavior), not for a learning use. The PG structure complements gradient-based attribution and addresses limitations of existing reasoning-interpretability methods. In contrast, RPG [1] and EM-PG [2] develop new policy-gradient algorithms aimed at improving model performance through fine-tuning.
>
> >**Reference**
> [1] Zhang, Yifan, et al. "On the design of kl-regularized policy gradient algorithms for llm reasoning." arXiv preprint arXiv:2505.17508 (2025).
> [2] Xu, Tianbing. "Training Large Language Models to Reason via EM Policy Gradient." arXiv preprint arXiv:2504.18587 (2025).
> [3] Højer, Bertram, Oliver Jarvis, and Stefan Heinrich. "Improving reasoning performance in large language models via representation engineering." arXiv preprint arXiv:2504.19483 (2025).
> [4] Rai, Daking, and Ziyu Yao. "An investigation of neuron activation as a unified lens to explain chain-of-thought eliciting arithmetic reasoning of llms." arXiv preprint arXiv:2406.12288 (2024).
> [5] Sundararajan, Mukund, Ankur Taly, and Qiqi Yan. "Axiomatic attribution for deep networks." International conference on machine learning. PMLR, 2017.
> [6] Mueller, Aaron, et al. "The Quest for the Right Mediator: Surveying Mechanistic Interpretability for NLP Through the Lens of Causal Mediation Analysis." Computational Linguistics (2025): 1-48.

---

> ### Comment · Reviewer_2ZEq · 2025-11-23
> **Recommend this paper to be accepted.**
>
> Dear Authors,
>
> Thanks to the authors for their explanation, I have also benefited a lot. From my point of view, I am happy to see this paper being hired after reinforcement. I would like to recommend this paper to AC, SAC, etc.
>
> At the same time, I am willing to revise my evaluation to accept the proposal. Since I am not strictly a peer researcher in this subdivision field, I do not take a stand against oral or poster.
>
> Best wishes,

---

> > ### Author Response · Authors · 2025-11-24
> > **Author Response to Reviewer 2ZEq**
> >
> > Dear Reviewer 2ZEq,
> >
> > Thank you very much for your positive follow-up and for your willingness to revise your evaluation. We are glad to hear that our clarifications were helpful, and we are grateful for your recommendation to the AC/SAC. Your constructive feedback has significantly improved the quality of our paper.
> >
> > Thank you again for your time and effort in reviewing our submission.
> >
> > Best regards,
> > The authors of Paper 15897

---

### Official Review · Reviewer_JNjR · 2025-10-31

**Soundness:** 1
**Presentation:** 3
**Contribution:** 2
**Rating:** 2
**Confidence:** 4

**Summary:**

This paper provides a framework for attributing behaviors to internal components and then steering those components to control behavior. The paper is pretty well written and the methods are clear. However, the main problem is that this paper is not properly situated in the existing literature and fails to compare many baseline methods for model control. It also makes some very wrong claims about the state of existing literature.

The framing that mechanistic interpretability lacks causality is simply wrong. In fact, you cite Stolfo et al. (2023), a causal mediation analysis paper, in the sentence that makes that claim!

There is loads of work in the frameworks of causal mediation and abstraction analysis of LLMs. For example, https://arxiv.org/abs/2004.12265 and https://arxiv.org/abs/2004.12265; see https://arxiv.org/abs/2301.04709 or https://arxiv.org/pdf/2408.01416 for surveys with citations.

Steering literature. There really is a huge amount of steering literature and this is all about causality, i.e., causing the behavior to change. For example, https://arxiv.org/abs/2205.05124, or https://arxiv.org/pdf/2310.06824 or https://arxiv.org/abs/2306.0334 or https://arxiv.org/abs/2308.10248

You also should be aware of representation fine-tuning: https://arxiv.org/abs/2404.03592. This is should be included as a baseline in your paper. Also, axbench https://arxiv.org/abs/2501.17148 evaluates a bunch of methods for control, and you could evaluate your method on this benchmark to get standardized comparisons against a lot of different methods.


Without further experiments that demonstrate the performance of the method in context of existing literature, its hard to know whether it marks a novel improvement.

I think the focus on long term dependencies is a good part of the paper, but you would need to compare against something like ReFT with a language modeling loss to show you are better at handling long term dependencies. Overall, this method might be an improvement, but we can't know with the given set of experiments.

**Strengths:**

See summary

**Weaknesses:**

See summary

**Questions:**

See summary

---

> ### Author Response · Authors · 2025-11-22
> **Rebuttal by Authors**
>
> Dear reviewer JNjR, thank you for your comments and following is our response.
> > The main problem is that this paper is not properly situated in the existing literature and fails to compare many baseline methods for model control. It also makes some very wrong claims about the state of existing literature.
> The framing that mechanistic interpretability lacks causality is simply wrong. In fact, you cite Stolfo et al. (2023), a causal mediation analysis paper, in the sentence that makes that claim!
> There is loads of work in the frameworks of causal mediation and abstraction analysis of LLMs. For example, https://arxiv.org/abs/2004.12265 and https://arxiv.org/abs/2004.12265; see https://arxiv.org/abs/2301.04709 or https://arxiv.org/pdf/2408.01416 for surveys with citations.
> Steering literature. There really is a huge amount of steering literature and this is all about causality, i.e., causing the behavior to change. For example, https://arxiv.org/abs/2205.05124, or https://arxiv.org/pdf/2310.06824 or https://arxiv.org/abs/2306.0334 or https://arxiv.org/abs/2308.10248
>
> **(1) Response to the comment on "causality" and our position in existing literature**
> We thank the reviewer for the thoughtful comments and constructive criticism on our work. We acknowledge that our original phrasing around "causality" in the paper was not sufficiently precise and we would like to clarify both (i) the scope and main topic of our work and (ii) how IPG aligns with and extends existing causal mechanistic interpretability (MI) frameworks.
> * (i) For the scope of our contribution, our paper is not intended as a generic attribution with steering framework. The central research topic of our work is the reasoning mechanism of LLMs, and IPG is specifically designed to identify reasoning-critical components.
> * (ii) For our position in existing MI work,
>     - **Causality has always been important within mechanistic interpretability.** We fully agree with you that causality is always a core principle in much of the MI work. We apologize for our unclear phrasing and it can indeed be read as suggesting that MI lacks causality in general, which is not our intent. We have revised the "Introduction" in our paper to clearly acknowledge existing causal frameworks and to avoid this misunderstanding.
>     - **Gap Exists in causal analysis for complex reasoning.** While causal approaches exists (e.g., Stolfo et al. [1]), most causal MI methods so far are primarily focus on specific subskills of reasoning and shorter-range internal effects (such as simple arithmetic in [1]) of large language models (LLMs). There is a lack of causal frameworks specifically adapted for complex, multi-step reasoning tasks. How the internal components work within long-horizon Chain-of-Thought (CoT) is still opaque. Unlike previous methods that target specific tokens (e.g. arithmetic symbols), our approach addresses the challenge of analyzing the complete CoT generation process. IPG fills this gap by introducing (integrated) policy gradient and extends causal analysis to capture the long-horizon dependencies inherent in this process.
>     - **IPG can be viewed as a CMA-style method.** We thank the reviewer for directing us to the CMA survey [2]. IPG aligns perfectly with the CMA paradigm: we treat neuron/SAE feature sets as high-dimensional mediators and quantify their causal contributions to the final reasoning outcome via systematic interventions. This places IPG as a CMA-style, causality-aware approach tailored to the reasoning behavior of LLMs.
>     - **Relation to other steering literature.** We agree that the steering literature is fundamentally about causally changing model behavior. However, the primary goal in our work is not to propose a new steering technique but to understand the causal reasoning mechanism (i.e. which reasoning components causally mediate reasoning behavior). To this end, we adopt standard intervention methods shared across prior work [3,4,5,6]. Besides, we want to mention that methods based on contrastive prompts and Difference-in-Means vectors [7,8] might be effective for behavioral control. However, they are essentially input-centric: they manually craft prompts to elicit or suppress target behaviors, and the resulting steering vectors provide indirect evidence about the underlying mechanisms.  In contrast, IPG explicitly treats neuron or SAE feature sets as mediators and intervenes on them, aiming for structural causal attribution of the reasoning mechanism rather than only behavioral steering.
>
> In summary, we position IPG as a CMA-style, causality-aware approach specifically designed to handle *long-horizon, multi-step reasoning* of LLMs, while most existing CMA applications have focused on narrower behaviors (e.g., arithmetic) and where input-centric approaches struggle to provide structural causal explanations. We have revised the paper including the "Introduction" and "Related Work" discussions. Changes has been highlighted in blue.

---

> ### Author Response · Authors · 2025-11-22
> **Rebuttal by Authors**
>
> > You also should be aware of representation fine-tuning: https://arxiv.org/abs/2404.03592. This is should be included as a baseline in your paper. Also, axbench https://arxiv.org/abs/2501.17148 evaluates a bunch of methods for control, and you could evaluate your method on this benchmark to get standardized comparisons against a lot of different methods.
> Without further experiments that demonstrate the performance of the method in context of existing literature, its hard to know whether it marks a novel improvement.
>
> **(2) Response to Comment on added benchmark and baselines**
> Thanks for pointing out the need for additional baselines and benchmarks to contextualize our method's performance. In parallel to these baselines, we want to emphasize that the core contribution is proposing a new framework that can causally attribute the complex reasoning behaviors to model's inner working. *The causal attribution of long-term, multi-step reasoning* is a largely unexplored area where no existing benchmark is specifically designed for this setting. With this clarified, we now discuss how additional baselines and benchmarks, including AxBench and representation fine-tuning, relate to our setting.
>
> (i) **AxBench:**
> AxBench [10] focuses on *semantic concept control*, evaluating whether methods can manipulate static representation attributes (e.g., sentiment, style, occupation). This represents a mismatch with our focus on *long-horizon reasoning mechanisms*, which is a multi-step sequential decision making problem, not semantic attribute editing. Since AxBench does not contain reasoning tasks nor evaluate causal contributions to reasoning, it provides only a partial view of IPG. Our primary evaluation thus relies on downstream reasoning benchmarks (GSM8K, AIME, Math500), which directly measure the effect of intervening on reasoning-critical components.
> Nevertheless, we fully agree that AxBench offers a valuable, standardized testbed for MI and model control. Thus we have started running IPG on AxBench and **will report these results as they become available during the discussion period.**
>
> (ii) **ReFT:**
> We thank the suggestion for including ReFT [11] as a baseline. Per your suggestion, we have implemented ReFT as a baseline. **As shown in Table 3, IPG consistently outperforms ReFT on both models under our setting.**  ReFT is a powerful *training-based* method for controlling model behavior by optimizing a language modeling loss. Our IPG framework is conceptually different: IPG is a *training-free* framework and explicitly designed to identify reasoning-critical components and intervene on them to study causal effects on reasoning performance.
>
> **Experiment Setup:** To ensure a fair comparison, we trained ReFT on the same single layer (Layer 22 for Qwen model, Layer 24 for Llama model) using 1,000 samples from GSM8K train split. We used the best hyperparameter configuration for ReFT. We adopt the same evaluation settings in our paper (greedy decoding, zero-shot on GSM8K test set) and report only enhancement results as ReFT does not support suppression experiment.
>
> **Analysis:** Notably, ReFT leads to a *performance drop* under our setting. Such a phenomenon has also been observed in some other concurrent work [12]. We hypothesize two reasons for this degradation.
>
> - **ReFT loss optimizes languagem modeling, not reasoning.** ReFT finetunes the representation by minizing the language modeling loss, which mainly encourage the model to adapt to GSM8K's answer style rather than improving its internal reasoning ability. By contrast, IPG leverages a reward signal that directly evaluates the 'quality' of reasoning trace. Therefore, IPG can causally identify the true reasoning-critical components, *yielding consistent performance gains* on model even if it has already highly optimized on reasoning datasets.
> - **Overfitting on training data.** Both Qwen2.5-Math-1.5B-Instruct and Llama-3.1-8B-Instruct used in our paper are already fully optimized for mathematical reasoning, so any further training may leads to an overfitting to particular answer style.
> - **Data inefficiency of training-based method.** For a fair comparison with IPG, we train ReFT on only one layer using 1,000 samples, which is insufficient for a training-based method. Such limitation further highlights IPG's *data efficiency*.
>
> Table 3: Comparison of ReFT and IPG's accuracy (%) on GSM8K
> | Models |Original|IPG (Enhanced)|ReFT|
> | --- | --- | --- | --- |
> | Qwen2.5-Math1.5B-Instruct|82.41 | 84.38 |62.47|
> | Llama3.1-8B-Instruct |85.89 |87.41 |77.94|

---

> > ### Author Response · Authors · 2025-12-03
> > **Further discussion on AxBench**
> >
> > **Preliminary AxBench [1] results:**  In our earlier response, we committed to evaluating our IPG method on AxBench. Due to the coming end of the discussion period, we can no longer have further exploration on AxBench. We will report our preliminary results here for completeness and to honor that commitment. Our IPG yields **moderate but consistent gains in concept-steering score over the baseline method like PCA, LAT** from the paper [1], while leaving the fluency unaffected.
> >
> > **Experiment:** We implemented a **minimal AxBench adaptation of IPG** that **directly intervenes on the base model’s hidden states without any additional representation learning (e.g., SAEs) and careful design of our reward function**. We ran attribution on the *CONCEPT 10* dataset on the 10th layer (L10) of Gemma2-2B [2]. We chose the occurrence of the pre-defined concept-related words as our reward function for IPG attribution. The result has been shown in the table below.
> >
> > **Analysis:** Despite the time constraint and basic implementation, IPG still delivers **consistent positive steering effects** over classical baselines (PCA, LAT, Probe). Moreover, our IPG can be further integrated with SAE features like that in our paper on LLM reasoning tasks, which we expect will enhance its concept steering performance substantially.
> >
> > On top of it, we note that IPG is basically designed for interpreting long-term, multi-step reasoning, not concept editing. Therefore, AxBench provides a partial view of IPG from semantic concept editing and we mainly regard the experiments as a **sanity check demonstrating that IPG can transfer to AxBench-style concept-based model steering**. A comprehensive study on AxBench that combines IPG with SAE features with more carefully designed reward signals (like a well-trained concept classifer) is left for future work.
> >
> > We thank again the reviewer for their valuable comments and suggestions.
> >
> > Table: Rank-1 steering leaderboard on concept steering.
> >
> > | Method | Catgory | Gemma2-2B L10 |
> > | --- | --- | --- |
> > | SAE | SAE feature | 0.177 |
> > | SAE-A | SAE feature | 0.166 |
> > | IPG (Neuron) | Non-SAE | 0.128 |
> > | LAT | Non-SAE | 0.117 |
> > | PCA | Non-SAE | 0.107 |
> > | Probe | Non-SAE | 0.095 |
> >
> > [1] Wu, Zhengxuan, et al. "AxBench: Steering LLMs? Even Simple Baselines Outperform Sparse Autoencoders." Forty-second International Conference on Machine Learning.
> >
> > [2] Team, Gemma, et al. "Gemma 2: Improving open language models at a practical size." arXiv preprint arXiv:2408.00118 (2024).

---

> ### Author Response · Authors · 2025-11-22
> **Rebuttal by Authors**
>
> > I think the focus on long term dependencies is a good part of the paper, but you would need to compare against something like ReFT with a language modeling loss to show you are better at handling long term dependencies. Overall, this method might be an improvement, but we can't know with the given set of experiments.
>
>
> **(3) Discussion on why IPG better handling long-term dependencies**
> From Table 3 we can see that ReFT [11] has inferior performance on handling long-term dependancies.
>
> (i) **Policy gradient style attribution allows to aggregates long-term information.** In multi-step reasoning, early computations often impose a cumulative influence on the final answer that local supervision cannot capture. IPG captures this by linking the model's final outcome signal (i.e. outcome reward) with the internal activation at each step of reasoning process. This allows IPG to better attribute reasoning critical components which are pivotal for the entire reasoning chain. Meanwhile, prior work like Deepseek-R1 [7] has further proved that the policy gradient-based method better aligns with reasoning quality than language modeling loss. However, IPG uses this principle for component attribution and control, not for training, giving a new contribution beyond reward-based optimization.
>
> (ii) **Integrated gradient allows reliable causal attribution.** With integrating gradients accumulating along a path from baseline to actual activation, we achieve completeness and faithfulness in causally attributing reasoning behavior. This paves the way for intervening on the identified components to show our controllability.
>
> >**Reference**
> [1] Alessandro Stolfo, Yonatan Belinkov, and Mrinmaya Sachan. 2023. A Mechanistic Interpretation of Arithmetic Reasoning in Language Models using Causal Mediation Analysis. In *Proceedings of the 2023 Conference on Empirical Methods in Natural Language Processing*, pages 7035–7052, Singapore. Association for Computational Linguistics.
> [2] Mueller, A., "The Quest for the Right Mediator: Surveying Mechanistic Interpretability Through the Lens of Causal Mediation Analysis", <i>arXiv e-prints</i>, Art. no. arXiv:2408.01416, 2024. doi:10.48550/arXiv.2408.01416.
> [3] Dai, Damai, et al. "Knowledge neurons in pretrained transformers." Proceedings of the 60th Annual Meeting of the Association for Computational Linguistics (Volume 1: Long Papers). 2022.
> [4] Shi, Yingdong, et al. "Dissecting and mitigating diffusion bias via mechanistic interpretability." Proceedings of the Computer Vision and Pattern Recognition Conference. 2025.
> [5] Ferrando, Javier, et al. "Do I Know This Entity? Knowledge Awareness and Hallucinations in Language Models." The Thirteenth International Conference on Learning Representations.
> [6] Mengru Wang, Ziwen Xu, Shengyu Mao, Shumin Deng, Zhaopeng Tu, Huajun Chen, and Ningyu Zhang. 2025. Beyond Prompt Engineering: Robust Behavior Control in LLMs via Steering Target Atoms. In Proceedings of the 63rd Annual Meeting of the Association for Computational Linguistics (Volume 1: Long Papers).
> [7] Marks, S. and Tegmark, M., "The Geometry of Truth: Emergent Linear Structure in Large Language Model Representations of True/False Datasets", <i>arXiv e-prints</i>, Art. no. arXiv:2310.06824, 2023. doi:10.48550/arXiv.2310.06824.
> [8]Turner, A. M., "Steering Language Models With Activation Engineering", <i>arXiv e-prints</i>, Art. no. arXiv:2308.10248, 2023. doi:10.48550/arXiv.2308.10248.
> [9] DeepSeek-AI, "DeepSeek-R1: Incentivizing Reasoning Capability in LLMs via Reinforcement Learning", <i>arXiv e-prints</i>, Art. no. arXiv:2501.12948, 2025. doi:10.48550/arXiv.2501.12948.
> [10] Wu, Zhengxuan, et al. "AxBench: Steering LLMs? Even Simple Baselines Outperform Sparse Autoencoders." Forty-second International Conference on Machine Learning.
> [11] Wu, Zhengxuan, et al. "Reft: Representation finetuning for language models." Advances in Neural Information Processing Systems 37 (2024): 63908-63962.
> [12] Liang, S., Cao, P., Zhao, J., Huang, C., Zhao, J., and Liu, K., "Bias-Restrained Prefix Representation Finetuning for Mathematical Reasoning", <i>arXiv e-prints</i>, Art. no. arXiv:2511.10707, 2025. doi:10.48550/arXiv.2511.10707.

---

### Official Review · Reviewer_dx8d · 2025-11-01

**Soundness:** 2
**Presentation:** 2
**Contribution:** 2
**Rating:** 6
**Confidence:** 2

**Summary:**

This paper introduces integrated policy gradient (IPG), a gradient-based method that attributes intermediate activations to some final outcome. This is done as follows:
1. Take the gradient $\nabla_h E_\pi[J(h)]$  where $J$ is the outcome (reward). Do so using the score function / policy gradient trick to get unbiased estimates.
2. Estimate this individually for each component of some hidden state, and average over all vectors from the latent vector $h_i$ to some baseline $h_i’$ (path gradient to reduce noise)
3. Pick the top p components by the average magnitude of their IPG over some dataset

After identifying the top components, the behaviour of the LLM can be controlled via scaling each component by a scalar to be larger / smaller. The authors suggest it is better to do this in k-sparse autoencoder latent space, rather than over raw neurons. The authors find that IPG can steer vectors to drive LM behaviour in the form of improved output accuracy, and do so better than other baseline interpretability methods.

**Strengths:**

The use of the policy gradient / score function trick to estimate gradient-based attribution is very clever. This was thought provoking for me to think about how PG can be applied generally to interpretability. I think this is both original and potentially significant.

The writing is generally easy to follow. The motivation to move beyond correlation-based methods is justified, and the main claims seem supported.

**Weaknesses:**

It may be due to my lack of expertise in interpretability, but there are some problem set-up and design choices I do not fully understand. I discuss them in the questions below.

It would also be nice to have not only math and reasoning performance, but steering for harmfulness, for example.

Formatting issues: L246-247, figure is blocking main text (likely due to negative vspace)

**Questions:**

**Q1:** How is the advantage estimated in Eq 1? L293 seems to suggest it is the GRPO advantage, but L164/165 mentions “accuracy reward”? Appendix A.3 also mentions IPG variants and GRPO advantage.

For what it’s worth, the GRPO advantage is a biased estimate of the true policy gradient (particularly the part where it divides the advantage by its standard deviation). I mention this because it feels like while having low-variance, biased gradients may work well in the training case (where GRPO is applied), it feels very important to have un-biased gradients in the interpretability case so you can accurately estimate contribution. You can always reduce variance by sampling and averaging over larger batches. It would be nice to see a discussion / ablation on this.

**Q2:** On controlling behaviour via scaling activations (Eq 4), does this assume some kind of linearity in the representation? I.e. increasing activation magnitude always increases some kind of output behaviour? Is the idea that in order to do good interpretability we must find components that obey this relationship?

**Q3** Architecture assumptions. Section 4.5 mentions steering DeepSeek-R1-
Distilled-Qwen-1.5B using neurons identified in Qwen2.5-Math-1.5B-Instruct. Generally speaking, this can only be done for models with identical architecture, vocabulary, and tokenizer, right?

**Q4** It is not clear to me what argument Figure 4 is making. For instance in Fig 4b, what should we read from the Jaccard similarity numbers? IPG’s _lower_ Jaccard similarity is argued as “more meaningful”, which I am not sure I follow. Wouldn’t higher Act similarity across the math datasets indicate better generalization of discovered components?

---

> ### Author Response · Authors · 2025-11-22
> **Rebuttal by Authors**
>
> Dear reviewer dx8d, thank you for your appreciation and comments. Following is our response.
> > W1. It would also be nice to have not only math and reasoning performance, but steering for harmfulness, for example.
>
> Thank you for the suggestion. While our paper primarily focuses on reasoning, our method is not restricted to this. IPG can also be applied to other domains besides math or reasoning, such as safety. To demonstrate its broader applicability, we additionally evaluate IPG on the **WildGuard**, a harmfulness benchmark for language models [1]. The results suggest that our **IPG generalizes well to the safety domains**. The results are shown in Table 1 below.
>
> **Experiment Settings:** For each prompt, we generate responses *before and after intervention*, and classify each response as *harmful* or *non-harmful* using the WildGuard model [1]. Our evaluation metric is:  $\text{harmlessness rate} =\frac{\\# \text{non-harmful responses}}{\\# \text{total responses}}$. A higher harmlessness rate means the model produces fewer harmful outputs, and is therefore safer. Activation method is adopted as the baseline. Consistent with our paper, it selects neurons with the highest activation values. As in Table 1 of the paper, we use greedy decoding for all generations. "Enhance" strengthens components that make the model less harmful (higher is better), while "Suppress" weakens those components (lower is better).
>
> **Analysis:** Table 1 shows that IPG effectively identifies the internal components that *causally* attribute to the harmful behavior, resulting in a controllable and effective steering.
>
> Table 1: Comparison of IPG and baseline method on controlling harmfulness for Qwen2.5-1.5B-Instruct.
> |Method | Original (%) | Enhance $\uparrow$ (%)| Suppress $\downarrow$ (%)|
> |-------|-------|-------|-------|
> | IPG (ours) | 89.52 |90.27| 85.81|
> | Activation | 89.52 |89.64| 89.05|
>
> > Q1.1 How is the advantage estimated in Eq 1? L293 seems to suggest it is the GRPO advantage, but L164/165 mentions "accuracy reward"? Appendix A.3 also mentions IPG variants and GRPO advantage.
>
> Sorry for causing the confusion. We have revised the content in L174.
> * The advantage in Eq. (1) is estimated following GRPO method [2], which normalizes the gained reward $r_t$ at the reasoning step $t$ using the mean value and standard deviation of the rewards from a group of trajectory samples, as is discussed in Eq. (7) in Appendix A.3 of our paper.
> * The "accuracy reward" mentioned in L164/165 means that the implementation of reward $r_t$ used in the advantage function is the final answer correctness.
> In general, this reward can take various formulations, such as (i) rule-based rewards [3] calculated by a verifier function on assessing the correctness of the final response following the LLM reasoning; (ii) model-based rewards [3] utilizing a reward model to estimate the return of the LLM response. There're more discussions about these reward variants in Appendix A.3.

---

> > ### Author Response · Authors · 2025-11-22
> > **Rebuttal by Authors**
> >
> > > Q1.2 The GRPO advantage is a biased estimate of the true policy gradient,it feels very important to have un-biased gradients in the interpretability case so you can accurately estimate contribution. You can always reduce variance by sampling and averaging over larger batches. It would be nice to see a discussion / ablation on this.
> >
> > Thanks for pointing out the variance problem. The normalization part in GRPO advantage can indeed introduce variance, as stated in [4]. GRPO treats the standardized reward (i.e., divided by its standard deviation) as its advantage. When the group size and batch size are small, the estimated advantage might be unstable, leading to a biased gradient with high variance. We believe this is what you are referring to.
> >
> > **Our empirical results indicate that the effect of GRPO's bias is minor and our method is robust to this bias when we use reasonable batch sizes.** We derive some empirical analysis as below.
> >
> > For illustration, we conduct an ablation on batch size (number of samples) below from 1 to 500, and evaluate how it influences the controlling effects on reason behavior on the downstream task. As the number of samples increases, the variance is consistently reduced, resulting in a more stable and less biased policy gradient estimate.
> >
> > **Experiment Settings:** We adopt the same setup as Table 1 in our paper: Qwen2.5-Math-1.5B-Instruct evaluated on GSM8K with greedy decoding. "Enhance" measures the accuracy after strengthening the identified components (higher is better), while "Suppress" measures accuracy after weakening them (lower is better). The original model's performance is 82.41%. The result shows that
> > - When batch size is small ($\le$ 10), the gradient signal is under high-variance. This may lead to an inaccurate detection of reasoning components, weakening the intervention effect (i.e., poor suppressing).
> > - As the batch size increases ($\ge$ 20), the biased gradient is smoothed and the gradient becomes more reliable, leading to more stable intervention performance and substantially improved suppression and enhancement.
> >
> >
> >
> > Table 2: Effect of batch size on steering performance.
> > | # Samples | Enhance $\uparrow$ | Suppress $\downarrow$|
> > | --- | --- | --- |
> > | 1 | 82.79 | 82.03 |
> > | 10 | 83.17 | 79.37 |
> > | 20 | 83.17 | 06.52 |
> > | 50 | 83.54 | 12.81 |
> > | 100 | 83.39 | 12.81 |
> > |500|83.55 |12.05 |
> >
> > We have revised the paper according to your comments, including (i) refining the description of advantage function in Appendix A.3 and (ii) adding the experiment and discussion of the batch size hyperparameter in Appendix D.1.
> >
> > > Q2.  Does controlling behaviour via scaling activations (Eq 4) assume some kind of linearity in the representation? I.e. increasing activation magnitude always increases some kind of output behaviour? Is the idea that in order to do good interpretability we must find components that obey this relationship?
> >
> > We make clarifications for "scaling" from the two aspects below.
> >
> > (i) **Scaling activations does rely on a linearity assumption.** The hypothesis underlying "scaling" is that if we amplify the activation magnitude, it should have stronger effects on target behavior, which is common practice in mechanistic interpretability works [5,6,7,8].
> >
> > (ii) **Good interpretability does not necessarily mean we must find components that obey this linearity relationship.** Instead, it should focus on their contribution to target behavior or concept. While "scaling" implicitly favors components with linear influences, interpretability does not hinge on this. Instead, it prioritizes components that drive target behavior. Our IPG method first quantifies how they contribute to target behavior, while scaling just serves as an effective method to show the contribution of the identified components to the model's target behavior, not a prerequisite for discovery.
> >
> > However, we strongly believe that developing more effective controlling methods, beyond simply scaling, based on the interpretability results of our method (or other mechanistic interpretability works) that can fully exert the effects of identified components on model behavior can be worthy of future exploration. Thank you for your valuable comments.

---

> ### Author Response · Authors · 2025-11-22
> **Rebuttal by Authors**
>
> > Q3. Architecture assumptions. Section 4.5 mentions steering DeepSeek-R1- Distilled-Qwen-1.5B using neurons identified in Qwen2.5-Math-1.5B-Instruct. Generally speaking, this can only be done for models with identical architecture, vocabulary, and tokenizer, right?
>
> Thank you for highlighting this point.
>
> (i）**Yes, it does assume the same architecture**. The transfer experiment in Section 4.5 is to study how reasoning-related components behave across closely related models, including reasoning-distilled variants (e.g. Qwen2.5-Math1.5B and Deepseek-distilled Qwen1.5B). It does assume that the source and target models share the same architecture, for example, the same hidden dimensionality, so that the neuron (or feature) indices are corresponded across models. This is necessary because the intervention is applied to the same component positions in both models.
>
> (ii) **Architectural alignment is not a requirement of IPG itself**, but only of the optional cross-model transfer experiment. This requirement applies only to the "transfer" setting. IPG itself does *not* require architectural alignment; it can be applied to any model individually. The assumption is needed solely when one wants to reuse discovered components across models. In practice, this setting is quite natural for families, such as a base model and its distilled versions, where architectures are matched.

---

> ### Author Response · Authors · 2025-11-22
> **Rebuttal by Authors**
>
> > Q4. It is not clear what argument Figure 4 is making. For instance in Fig 4b, what should we read from the Jaccard similarity numbers? IPG’s lower Jaccard similarity is argued as "more meaningful", which I am not sure I follow. Wouldn’t higher Act similarity across the math datasets indicate better generalization of discovered components?
>
> Sorry for the ambiguity. Figure 4 shows the pair-wise Jaccard similarity between selected neuron sets among four benchmarks. Importantly, our argument is that, under this setting, a moderate level of similarity is desirable, not extremely high or extremely low because reasoning tasks share some common structure but also contain dataset-specific differences. We clarify this with two points below:
> * (i) **Activation's high Jaccard similarity reflects method bias, not genuine cross-task generalization.** The Activation baseline exhibits abnormally high similarity across benchmarks. However, this does not indicate the discovery of general reasoning components. Instead, it simply selects neurons with consistently large activation magnitudes, which is named "massive activation" phenomenon [9] where some neurons activate strongly regardless of input. As these neurons may not be task-specific, the Activation method fails to reliably control reasoning behavior (consistent with relatively low improvement accuracy shown in Table 1 in paper).
> * (ii) **IPG's moderate Jaccard similarity indicates that it captures shared reasoning structure while still adapting to dataset-specific reasoning demands.** IPG shows a more balanced pattern because the selected neurons overlap across benchmarks (higher than random selection) but remains far from identical (lower than Activation). This behavior matches our expectation. Reasoning datasets naturally share some underlying skills (e.g., arithmetic, multi-step planning), so partial overlap is expected. However, the datasets also differ in difficulty and reasoning style, so the critical components should not be exactly the same.
>
> We have revised the paper to be clearer in Section 4.3.
>
> >**Reference**
> [1] Han, Seungju, et al. "Wildguard: Open one-stop moderation tools for safety risks, jailbreaks, and refusals of llms." Advances in Neural Information Processing Systems 37 (2024): 8093-8131.
> [2] Shao, Zhihong, et al. "Deepseekmath: Pushing the limits of mathematical reasoning in open language models." arXiv preprint arXiv:2402.03300 (2024).
> [3] Guo, D., Yang, D., Zhang, H. et al. DeepSeek-R1 incentivizes reasoning in LLMs through reinforcement learning. Nature 645, 633–638 (2025). https://doi.org/10.1038/s41586-025-09422-z
> [4] Liu, Z., Chen, C., Li, W., Qi, P., Pang, T., Du, C., Lee, W. S., & Lin, M. (2025). Understanding R1-Zero-Like Training: A Critical Perspective. In 2nd AI for Math Workshop @ ICML 2025.
> [5] Dai, Damai, et al. "Knowledge neurons in pretrained transformers." Proceedings of the 60th Annual Meeting of the Association for Computational Linguistics (Volume 1: Long Papers). 2022.
> [6] Shi, Yingdong, et al. "Dissecting and mitigating diffusion bias via mechanistic interpretability." Proceedings of the Computer Vision and Pattern Recognition Conference. 2025.
> [7] Ferrando, Javier, et al. "Do I Know This Entity? Knowledge Awareness and Hallucinations in Language Models." The Thirteenth International Conference on Learning Representations.
> [8] Mengru Wang, Ziwen Xu, Shengyu Mao, Shumin Deng, Zhaopeng Tu, Huajun Chen, and Ningyu Zhang. 2025. Beyond Prompt Engineering: Robust Behavior Control in LLMs via Steering Target Atoms. In Proceedings of the 63rd Annual Meeting of the Association for Computational Linguistics (Volume 1: Long Papers).
> [9] Sun, Mingjie, et al. "Massive Activations in Large Language Models." First Conference on Language Modeling.

---

### Meta-Review · Area_Chair_JBr5 · 2026-01-13

**Summary:**

This paper proposes a training-free framework for interpreting and controlling reasoning behavior in large language models. The method uses gradient-based attribution combined with policy gradients to identify internal components that contribute to reasoning outcomes. The paper present a way to propagate outcome-based reward signals backward through inference trajectories for attribution and includes experiments on benchmark datasets.

This paper received 3 reviews with mixed feedback from reviewers with mixed expertise (6, 4, 2). Overall, reviewers acknowledged the method studies an important topic (locating reasoning features inside LLMs) and found the use of policy gradients for attribution interesting. On the other hand, one reviewer – JNjR raised concerns about the significance and soundness of the contribution – particularly with respect to existing causal interpretability literature, including causal mediation analysis and steering methods.

This paper received 3 reviews with mixed feedback (overall scores 6, 4, 2; confidence 2, 3, 4 for dx8d, 2ZEq, JNjR). In this case, the feedback from reviewers was mixed. On the one hand, all reviewers found the topic timely and the proposed attribution method interesting, and they viewed the intervention results as suggestive (dx8d, 2ZEq, JNjR). On the other hand, two of them raised concerns about positioning, baselines, and whether the evidence supports the stronger claims, especially around causality and mechanistic interpretability (2ZEq, JNjR).

Having read the reviews and the paper, I am unfortunately recommending rejection at this stage. My decision is motivated by feedback JNjR, which I weigh more heavily given their higher confidence. JNjR argues that the paper is not well situated in existing literature and that parts of the causality framing are incorrect or overstated without a careful treatment of prior work. JNjR and 2ZEq also ask for stronger comparisons to relevant control and steering baselines, and more analysis to support causal claims rather than correlational attribution (2ZEq, JNjR). Overall, the paper would benefit from a tighter scope and more precise claims, supported by stronger baselines and targeted ablations.

**Reviewer Concerns:**

Outstanding
- Positioning (paper not properly situated within causal interpretability and steering literature) (JNjR)
- Claims about mechanistic interpretability lacking causality are inaccurate (JNjR)
- Missing Baselines (incomplete comparison against ReFT and standardized benchmarks like AxBench) (JNjR)

Resolved/Ignored
- Distinction from policy-gradient optimization methods (2ZEq)
- Limited Transfer Gains (cross-benchmark improvements are modest) (2ZEq)

**Reviewer Scores:**

I would bet on (6, 4, 2) -> (7, 6, 2) with JNjR retaining their score.

---

### Decision · Program_Chairs · 2026-01-26

Reject